# The US Dobson Station Network Data Record Prior to 2015, Re-evaluation of NDACC and WOUDC archived records with WinDobson processing software

Robert D. Evans[2], Irina Petropavlovskikh[1], Audra McClure-Begley[1], Glen McConville[1], Dorothy Quincy[2], Koji Miyagawa[3]

[1]Cooperative Institute for Research in Environmental Sciences, University of Colorado, Boulder 80309, USA

[2]Retired from NOAA/ESRL, Global Monitoring Division, Boulder, CO, 80305, USA

[3] Visitor with NOAA/ESRL, Global Monitoring Division, Boulder, CO, 80305, USA

*Correspondence to*: Robert D. Evans (Robert.D.Evans@noaa.gov)

**Abstract.** The United States government has operated Dobson Ozone Spectrophotometers at various sites, starting during the International Geophysical Year (July 1, 1957 to December 31, 1958). A network of stations for long-term monitoring of the total column content (thickness of the ozone layer) of the atmosphere was established in the early 1960s, and eventually grew to sixteen stations, fourteen of which are still operational and submit data to the United States of America's National Oceanic and Atmospheric Administration (NOAA). Seven of these sites are also part of the Network for the Detection of Atmospheric Composition Change (NDACC), an organization that maintains its own data archive. Due to recent changes in data processing software the entire data set was re-evaluated for possible changes. To evaluate and minimize potential changes caused by the new processing software, the reprocessed data record was compared to the original data record archived in the World Ozone and UV Data Center (WOUDC) in Toronto, Canada. The history of the observations at the individual stations, the instruments used for the NOAA network monitoring at the station, the method for reducing zenith sky observations to total ozone, and calibration procedures were re-evaluated using data quality control tools built into the new software. At the completion of the evaluation, the new data sets are to be published as an update to the WOUDC and NDACC archives, and the entire data set is to be made available to the scientific community. The procedure for reprocessing Dobson data and the results of the re-analysis on the archived record is presented in this paper. A summary of historical changes to fourteen station records is also provided.

The new WinDobson data is now available from ftp://aftp.cmdl.noaa.gov/data/ozwv/Dobson/WinDobson/. The WOUDC archive is available from http://woudc.org/, and the NDACC archive is available from ftp://ftp.cpc.ncep.noaa.gov/ndacc/. The WinDobson Software as extended by NOAA, Level-0 Dobson data and calibration records are available on request to NOAA Dobson network personnel (https://www.esrl.noaa.gov/gmd/ozwv/dobson/contact.html.)

## Background

The Dobson ozone spectrophotometer was designed in the 1920s and is still in use today. The instrument is fully described elsewhere (Dobson, 1931, 1968), but briefly, it measures the relative intensity of solar radiation between selected wavelength pairs in the range of 300-350 nanometers. These pairs are named A (305.5 and 325.4 nanometers (nm)), C (311.5 and 334.4 nm), C' (334.4 and 453.6 nm) and D (317.5 and 339.8 nm); and are combined in the measurement process either as A and D (AD); C and C' (CC'); or C and D (CD). The optical arrangement of the instrument is presented in Fig. 1. Measurements on either direct solar light or light scattered from the zenith can be used to calculate the amount of ozone between the instrument and the top of the atmosphere (total ozone column or TOC). Approximately 90% of this TOC resides in the region between 15 and 30 km above the Earth's surface that is defined as the ozone layer.

The relative intensity of wavelength pairs measured if instrument was operated outside the Earth's atmosphere is referred to as the extraterrestrial constant (ETC). The instrument's ETC is determined either through a Langley Plot method (Langley, 1884) or by direct comparison with a standard Dobson instrument (Komhyr and Evans, 2008). The concept of measurement of the Dobson spectrophotometer exploits the change in the ratio of Solar light intensities measured at the respective wavelength pairs as caused by the passage of UV light through the ozone layer. The light enters the instrument (Fig. 1), and the right side of the optics produce a spectrum projected on a slit arrangement containing slits $S_2$, $S_3$ and $S_4$ (only used for C' measurements). The left side of the optics combines the images of the slits into the photomultiplier tube. The measurement of the relative intensities is performed by moving a neutral density filter (the optical "wedge") across the light path of the wavelength less absorbed by ozone, specifically the light passing through slit $S_3$. The wedge is moved to reduce the light intensity of $S_3$ to equal the intensity of $S_2$, the wavelength more absorbed by ozone, as seen by the photomultiplier tube. The instrument's output, R-value, indicates the position of the neutral density filter as indicated on an engraved plate, which is the primary measurement and is specific to the Dobson instrument optical properties.

The N-value represents the attenuation by ozone and by other scattering and absorption during the UV light's passage through the atmosphere between the sun and the instrument. The relationship between R-values and N-values is both wavelength and instrument specific. R-values are converted to N values using tables called R to N tables (N-tables). These tables change during the instruments' lifespans due to repairs, updates and aging, thus each set has a limited period of application. The applicability of the N-table is monitored by means of intercomparisons with standard Dobson instruments, and with the use of instrument specific reference lamps. The calculation of ozone from observations made on the direct sun (DS) light (or reflected light from the moon) is with a defined algorithm based on Beer's law. The resolution of the measurement is 1 DU, and the precision (uncertainty) is considered to be ±1% (Grant, W,. 1989). Accuracy is another issue. The accuracy is dependent on knowledge of the ozone and temperature profile at the time of the measurement to correctly calculate the ozone absorption cross section. As this information is not available for individual observations, some assumptions must be made. A standard algorithm for the reduction of direct sun observations (Komhyr, et. al., 1993) is used by all organizations reporting daily values to the WOUDC or NDACC archives. The accuracy is also dependent on the knowledge of the ozone cross-section datasets used to determine the absorption coefficients in the reduction algorithm (Redondas, et al, 2014). The reduction of measurements on the zenith sky (ZS) is more complicated, as it is based on statistical analysis of DS and ZS observations close in time. The precision (uncertainty) of ZS empirical model is found to be 2-5% in this work, and is dependent on the wavelength pairs

used and the sky conditions. An accepted method of statistical analysis is not defined in the standard operating procedures; different organizations using the instrument employ different methods to build the empirical relation between direct sun and zenith sky measurements (Josefsson and Löfvenius, 2008.)

The Dobson instrument has limitations in the accuracy of measurements at certain observing conditions (Basher, 1982). Internal stray light is one such limitation. Moreover, each Dobson instrument has unique optical components that result in an instrument specific level of the stray light. The quality and aging stability of the individual wedge construction has improved over time; especially for instruments within the NOAA network, which had optical components replaced with those of a more robust design during instrument rebuilding in the 1980s.

Data reduction algorithms are fully discussed in the Section 7 of the Operations Handbook ( Komhyr & Evans 2008; http://www.wmo.int/pages/prog/arep/gaw/documents/GAW183-Dobson-WEB.pdf).

**Station History**

There are measurements of TOC in the USA prior to 1960 made by University and Federal organizations (Brönnimann, S., 2003), but the development of a coherent network of observing sites within the US Weather Service was started in the 1960s under the guidance of Walter Komhyr. The network was transferred to NOAA's Global Monitoring for Climate Change (GMCC) in the early 1970s, and is currently operated by NOAA's Earth System Research Laboratory's Global Monitoring Division (ESRL/GMD). As many as 16 stations comprised the network since its establishment. One station was closed; another was transferred to another parent authority. Table 1 displays the stations reporting at end of 2015. Originally, observations using the Dobson instruments were recorded with pen or pencil on forms designed to assist manual calculations (https://youtu.be/w1rV_96UChk). As computer power increased, the data was transcribed to punched cards for processing, then to direct entry by keyboard. By the mid-1990s, the NOAA instruments were equipped with computers and encoders, and the data was recorded in a "dayfile" at the time of the measurement. Six stations were equipped with fully automated instruments in the 1980s.

**Data Processing**

TOC is normally archived as a single representative value of TOC selected for each day. This not an average value, but the result from the "best" observation during the day. As the exact instrumentation and observational scheduling varies from station to station, the number of observations made daily also vary. The entire record of observations is available on request from NOAA. The full record of observations is available per request from NOAA Dobson network personnel listed at https://www.esrl.noaa.gov/gmd/ozwv/dobson/contact.html. In this publication, the term "select" value means the daily value produced in the NOAA processing stream. An earlier reprocessing of the stations' data was done in the 1990s (Komhyr, et al, 1995); the report also details much of the early history of measurements in the US system of stations.

To convert measurements to TOC values, calibration N-tables and information (reference lamp adjustment) from monthly instrument tests using reference lamps are required. The N-tables are defined by comparison of the station instrument to a reference standard. These referencings are normally done on a four to six year schedule. The calibration of the wedge is normally measured at the same time. The reference lamp tests are an indication

of the instrument's aging, but are only a single point test in the instrument measurement range. The comparison process measures instrument performance over a typical mu range of 1.15 to 3.85 but this is often adjusted with consideration to circumstances such as an instrument's location. The calibration N-tables are changed when the difference between the station and reference instrument is greater than the equivalent of 1% in TOC. When the calibration N-tables are changed due to a drift (determined from an inspection of the past calibrations, instrument operational history, and, if possible, comparison with other instrumental records), the existing data set from the last calibration change to the new calibration was reprocessed and re-published in the archives.

The set of computer programs used for the NOAA processing were written in the FORTRAN language, and by the 2010s were difficult to use and maintain due to changes in computer hardware and personnel. The decision was made to convert the NOAA processing to processing using the WinDobson software package, as the fully automated instruments were updated to a modern system based on this software. Developed by personnel of the Japan Meteorological Agency (Miyagawa, 1996), WinDobson is a software package for operations, data analysis and quality assurance of Dobson spectrophotometer observations. The algorithm for the reduction of ozone from DS observations with the Dobson is the standard method used by the NOAA software, but the ZS observations are reduced with a method described later in this manuscript. For the NOAA application, new components were developed. These new components are available from NOAA to other users of WinDobson. It is applicable for both TOC and Umkehr (ozone vertical profile) measurements. As this software has a different statistical method for the reduction of the zenith measurements, and set of rules (See section: Windobson Selection Rules) for determining the representative value of total ozone for each day with observations, the entire data record of each operational station was reprocessed in the WinDobson system to minimize the effect of the change when future data is placed in the archive  In the development of the data files and calibration information for Windobson processing, the entire record of observations, repair and calibration checks of each station was investigated and re-evaluated.   This investigation allows for correction of past errors.

**Data Format Conversion and Initial Comparison of Data Sets.**

The NOAA processed data were converted to "long line format" (LLF) files. These files are actually the image of the information sent to printers in the 1990s version of the data stream.  The select values for the WOUDC and NDACC archives were originally produced from these files, using a process of both machine and inspection by personnel. Programs were developed to convert the LLF and dayfiles into formats compatible with the WinDobson data stream. Files with instrument, station and calibration information (parafiles) were also developed to complete the structure of the WinDobson system. Connections to other sources of TOC information (satellite data records, for example) were developed so that comparisons with these values could be performed using tools internal to WinDobson.  Reference lamp values were extracted from the LLF records for time periods prior to 1995 and from the dayfiles afterwards.  By the end of 2015, all operational stations' data were being processed in WinDobson.

Initially, the data sets of only ADDS (fundamental wavelength pairs) observations from the two processing streams were compared with the expectation that the results should agree within ±1DU. The ADDS observations are considered the most reliable (fundamental), as the equation derived for conversion to ozone minimizes the

Rayleigh scattering term, and the aerosol term can be considered to be zero. Time periods with differences greater than this +/-1 DU were investigated to determine the source of the problem, and correct any differences. When the ADDS differences were reconciled, the ZS observations were compared to the DS observations to define a polynomial method within the WinDobson system for converting the ZS observations to TOC. Separate polynomials were defined for various time periods related to instrument repairs and calibration changes. The change in the methods of reduction of ZS measurements often produced large changes in reported TOC values. The improvement in the ZS results with respect to the ADDS results is displayed in Fig. 2 and in Table 3. The new method has resulted in ~91% of zenith sky derived total ozone (ADZB) within 2% of the coincident direct sun ozone column (ADDS). This is an improvement over the 78% value reported in the Operations Handbook (Komhyr and Evans, 2008). Results of observations made on the direct sun using the CD wavelength pairs differ from those made on AD pairs. The differences come primarily from imperfect knowledge of the ozone cross-sections used to determine the absorption coefficients used in the algorithm, and of the optical characteristics of the instrument (Redondas, et al, 2014.). The differences in observational results within a specific SZA range were analyzed, and a multiplying factor was established to bring the average of the CD results to that of the AD results with in the WinDobson system.

**Comparison of WinDobson Representative Values with Archived Daily Values**

The individual station records are archived as daily values in the World Ozone and Ultraviolet Radiation Data Centre (WOUDC) in Canada. (http://woudc.org/home.php). The format of reporting is a single value for the local day, but in UTC (Universal Time Coordinated) time with a resolution of an hour. The NDACC (http://www.ndsc.ncep.noaa.gov/data/) archive has the same TOC values for a subset of the WOUDC stations, but in a different format, with date and time in UTC. The reprocessed data sets will be archived in WOUDC and NDACC. For each station, tools in WinDobson were used to make a data set of daily representative TOC values. These data sets were compared to the data sets of select values downloaded from the WOUDC and NDACC, and the differences were investigated. The history of the instrument calibrations was again reviewed, and changes in the N-tables and the periods of the use of N-tables within the WinDobson system were made as needed. The differences stem from a number of reasons.

- There are data in the WOUDC data set for some stations that was reported by earlier organizations. The processing and selection rules for this data are unknown.
- The older (1995) processing included time periods of special processing to attempt to account for specific problems in the older optics of specific instruments. This was accomplished by a modification to the reference lamp correction used in the data processing. The lamp corrections for the pre-1995 processing were extracted from the LLF format and applied to the WinDobson data process to introduce the correction applied in the earlier processing. In some cases, the full correction was not possible to reproduce, so special N-tables were reconstructed from the information in the LLF format and applied on an annual basis. These periods are displayed graphically and discussed in the individual station reports. The problems that the special processing were attempting to correct were:
    - So-called Mu-dependency (Komhyr et al, 1995), where DS results are lower at low sun angles. As this effect is dependent on the intensity of the input solar beam, and thus on the TOC; no attempt was made to account for this effect in WinDobson processing. This problem is related to the internal scattered light in the instrument, which is difficult to evaluate.

- o Drifts in the shape of "wedge" calibration. It is unclear how the drift correction was actually performed in earlier processing; no attempt was made to account for this effect in WinDobson processing. Newer construction of the optical wedges used in the instrument have proved to have a much more stable calibration.
- o Drifts in the "extra-terrestrial constant" as part of the calibration. This was done in the WinDobson processing of later data, but with a different scheme -- multiple N-tables with shorter time periods of applicability.
- o There was a weakness in the NOAA processing in choosing a select value for each day. During the original review of observations, certain observations were rejected for selection; this rejection was not recorded in the LLF files, and thus rejected observations appeared in the WinDobson data set. We scrutinized the record for these discrepancies and amended the results.

- The results of the zenith measurements changed due to updates to the reduction method, and these type of changes affect all stations -- some of the changes are large and are discussed in the individual station reports.
- For some stations, it is common for observations to be made throughout the local day but with later observations being on the next consecutive UTC day. This occurs at Lauder (LDR), Samoa (SMO) and South Pole (SPO), where UTC date changes during normal observing period. For SPO, observations on a local day can differ by 22 hours; thus choice of the selected/representative ozone in the change from NOAA to WinDobson processing and selection may differ by 22 hours. At certain times of the year, the TOC can change appreciatively during this time period at SPO.
- Data archives sometimes failed to be updated after a calibration drift was detected during an intercomparison with a standard. This is not necessarily a failure of the internal WOUDC archiving process. NDACC appears to capture these periods more correctly.
- The rules for choosing the NOAA selected value for the day were similar to that of WinDobson, but were not consistent throughout the record or across stations, and the documentation of these rules is incomplete. For the WinDobson processing, the same rules are applied throughout the record and stations, and are described in the following section: Windobson Selection Rules.
- Our investigations of the station and instrument operation history revealed several periods for which different N-tables were used in the archived records as compared to the historical record of NOAA N-tables. Also, when a station instrument is compared to a standard instrument, and the results are within the uncertainty of the measurements (+/-1%), the station instrument's calibration is considered to be stable and thus is not changed. Otherwise, the instrument's calibration is changed and the existing data record starting from the time of the last comparison against a standard is reprocessed with the assumption that instrument's calibration has changed in a linear manner. Using the tools in WinDobson, our studies of the stations' records allowed comparisons with long term records indicating TOC. These comparisons showed that at certain stations, the calibration change was not linear. Further investigation of stations' history revealed damage to the instrument at that point (for example, rain entering the instrument shelter.) These investigations also identified instances where the comparison against the Dobson standard was not performed correctly, and therefore the calibration should not have been changed.

**Windobson Selection Rules.**

Often there are multiple observations on an individual day.  The observations are given an internal numeric code in WinDobson, based on the observation type, and operator input about the observation.  The representative value is chosen by the software with the priority groups given below, high to lowest. These groups are based on Table 2 in the Operations Handbook (Komhyr and Evans, 2008). If there are multiple observations of the highest priority on that day, the observation closest in time to local noon is chosen.  After the automatic selection, the daily representative values are reviewed by human inspection with possible intervention to select a different value.  The WinDobson software also has quality control routines that rates individual observations as good, questionable (flagged yellow) and likely bad (flagged red), based on internal consistencies of the measurements.  If an observation is rejected by the human inspector, the observation is not removed from the data record, but flagged as "not included".

Priority Groups are listed here; Operator inputs as to sky quality are included in determining priority:

1. Direct Sun observations using the AD pair combination with or without Ground Quartz Plate (diffuser) in the instrument's inlet window.  Observations with diffuser have higher priority.
2. Zenith Sky observations using the AD pair combination, observations on the clear zenith have higher priority over those on cloudy conditions
3. Direct Sun observations using the CD pair combination with Ground Quartz Plate (diffuser) in the instrument's inlet window.  Observations without diffuser have lower priority.
4. Zenith Sky observations using the CD pair combination, observations on the clear zenith have higher priority over those on cloudy conditions.
5. Zenith Sky  observations using the CC' pair combination, observations on the clear zenith have higher priority over those on cloudy conditions
6. Observations on light reflected from the moon. Observations using AD pair combination have higher priority.  Note these observations are rarely made other than at the South Pole Station during the austral winter.

**A discussion of the individual station records and the changes is presented in the following section.**

The station discussions are accompanied by a referenced graphic of the time dependent differences, consisting of either three panels (all stations) or five panels (NDACC) Stations. Panel A: The time record of total ozone measured at the station from the start of observations through 2014 (or until station was converted to WinDobson processing). Panel B: percent difference between daily WinDobson total ozone records compared to the WOUDC record, (WinDobson-WOUDC). The red line is a linear fit. Panel C: the same as the second but for monthly and yearly averages (based on all the values in the month in each data set). The small white circles are averages made from DS observations only; the red symbols represent averages using all Dobson total ozone records; the large black open circles are yearly averages of all observations, based on monthly averages. Large triangle symbols indicate major calibration or instrument changes that lead to creating the new N-tables; however, not all calibrations checks of the station record are shown as not all calibration checks revealed problems.  For NDACC Stations only: Panel D is the same as the second panel but for comparisons with the data archived at NDACC

center (WinDobson-NDACC). The black line is a linear fit. Panel E is the same as the third panel for comparisons with the NDACC archived monthly and yearly averages. NDACC values are not recorded as observation type. Table 2 displays standard statistics of the differences between WinDobson and WOUDC, and WinDobson and NDACC records.

Assessment of changes in the WinDobson representative dataset relative to WOUDC record is analyzed in the form of probability distributions, where percent differences in TOC are plotted (Fig 3) as function of likely change when the archive is updated. The datasets analyses are separated into ADDS and other type of measurements. The ADDS curves are symmetric, and indicate that the vast majority of ADDS values will be unchanged. The "other" curves are less symmetric, and are driven by the updated ZS reduction polynomials. As the overall record average offsets are small (<1.0%), this is an indication of the number of ADDS observations versus other observation types.

**Mauna Loa Observatory, Hawai'i, USA (19°N, 156°W, NDACC Station)**

Observations at MLO were started in December 1957. The instrument was damaged in 1961, and thus the calibration is unknown prior to 1963. Before 1984, the primary instrument was D063, with short periods with other instruments. The data in the archive prior to 1984 was not processed in the standard method in an attempt to account for instrument calibration drifts and other instrument problems, which causes larger variation in the comparison of original to the WinDobson record prior to 1984. The automated instrument D076 was installed at the station in 1984 after rebuilding in Boulder. A mirror deteriorated, so the calibration in the period 1990-1995 (indicated by the yearly N-table triangles) is based on comparisons with World Standard Dobson D083 while it was on station for Langley plot campaigns. (The Langley plot method is used to establish an Extra-terrestrial constant for an instrument (Langley, 1884).) This new calibration is not reflected in the WOUDC or NDACC archives. The instrument was rebuilt and the WinDobson automation installed in June, 2010. Data from 2010 through 2014 was processed in the NOAA system after converting WinDobson data files to a format compatible with the NOAA system. The NDACC archive appears to have updates not reflected in the WOUDC Archive, but there are periods with data missing from the NDACC archive (July December 2012). The difference between the WOUDC and NDACC archives records processed in the NOAA system and WinDobson system are presented graphically in Fig. 4.

**South Pole, Antarctica (90°S, 59°E, NDACC Station)**

South Pole Station was established in 1957. The first Dobson instrument failed due to the extreme cold. Observations started again in 1961 and these results are in the NOAA archive, but the calibration record dates from 1963. The normal routine established in 1985 is to change the instrument every four years for calibration checks, but this was not always achieved. This station has the possibility of large changes in reported daily values in the WinDobson, primarily due to the extended daily observation period, and high variation in total ozone during certain periods of the year. The station local day is the same as that of Christchurch, New Zealand for ease of logistics, but the Dobson observations are reported in the WOUDC in UTC date and hour. The date and time combination often is misleading (for example, In the WOUDC archive, 14 November 1994 has a time of 28 hours UTC, which matches the WinDobson and NDACC 15 November 1994 values.) The calculation of the astronomical parameters used in the algorithm for reducing reflected moon observations was incorrect in the

NOAA program throughout the period of record. Changes in the method of deriving total ozone from ZS observations improved the average with respect to DS averages, but creates differences between the old and new archives. There are several periods missing from the WOUDC and NDACC archives (for example, July through December 2002.). The difference between the WOUDC and NDACC archives records processed in the NOAA system and WinDobson system are presented graphically in Fig. 5. The exclusion of low TOC values in early October in the archived data (small white circles are outside of the plot range) in some years also produces large percentage differences in the averages (see large deviations in open circles seen in some years in the panel c and e). An example is October 1994, where there are 25 reported days in the WinDobson record but only 18 reported in the WOUDC, and only 10 in the NDACC archive. These inconsistencies can produce large percentage differences, especially during low ozone conditions.

The rules for selection and inclusion of days in the archives appear have been inconsistent in earlier (NOAA) processing and archiving. The NDACC archive prior to 1999 has TOC expressed as Vertical Column Density (molecules/cm**2). These numbers appear to have been calculated from DU, as this archive is derived from the WOUDC archive. There are periods where this calculation was done incorrectly (for example, October 1998, where the NDACC values differ by more than 100 DU when converted back to DU.) While the NDACC archive is supposed to be derived from the same internal NOAA archive as WOUDC, there are random differences (For example, February 1981 is missing from the NDACC archive.) The change in the yearly cycle of TOC (Panel A) is evident in the austral spring due the depletion related to chlorofluorocarbon release (Farman etal, 1985). Station and observing schedules were changed to accommodate research needs after that 1985.

**Bismarck, North Dakota, USA (47°N, 101°W)**

The instrument is operated by the National Weather Service office at Bismarck Airport. There are observations in the archive from the late 1950s, but the documented record starts in December 1962. The difference between the WOUDC and NDACC archives records processed in the NOAA system and WinDobson system are presented graphically in Fig. 6. The periods where the N-tables were reconstructed from the results of the special processing in 1995 are indicated by the yearly N-table triangles. The instrument's calibration has been quite stable since 1995.

**Caribou, Maine, USA (47°N, 68°W)**

The instrument is operated at the National Weather Service office at the Caribou Airport. There are observations in the archive from the late 1950s, but the documented record starts in August 1962. The Weather service office was rebuilt in the early 2000s, with data gaps during that period of the record. The difference between the WOUDC and NDACC archives records processed in the NOAA system and WinDobson system are presented graphically in Fig. 7. The periods where the N-tables were reconstructed from the results of the special processing in 1995 are indicated by the yearly N-table triangles. The instrument's calibration has been quite stable since 1995.

**Nashville, Tennessee, USA (36°N, 87°W)**

The instrument is operated at the National Weather Service office near Old Hickory, Tennessee. There are observations in the archive from the late 1950s, but the documented record starts in July 1962. This station record shows a larger offset (+0.6%) between the WOUDC and WinDobson data sets, due to the change to the zenith observations results. The difference between the WOUDC and NDACC archives records processed in the NOAA system and WinDobson system are presented graphically in Fig. 8. The periods where the N-tables were reconstructed from the results of the special processing in 1995 are indicated by the yearly N-table triangles.

**Fairbanks, Alaska, USA (65°N, 148°W)**

Observations were started at the Fairbanks airport in 1964 using instrument D076, but ceased in 1972. The values in the WOUDC archive in the 1964-1972 period do not correspond to the values in the older NOAA internal archive for reasons not determined. Observations were restarted at the Poker Flat Research Range (65°N, 147°W) in 1985. The mission of the Range changed in 1993 and the Dobson shelter was moved to the roof of the Geophysical Institute at the University of Fairbanks. Operations restarted in April 1994. This station is at 65 degrees north, with observations on low sun with high ozone amounts common, especially in March and April. Researchers are advised that this instrument shows patterns in the comparison with other instrumentation that imply an under estimation of ozone on the ADDS wavelength under conditions of low sun and high ozone. The older NOAA processing and selection of observations was different from other stations, as CD pair combinations were often selected over AD pair combinations, while WinDobson uses the same rules for all stations. This change in selection is reflected in the variability in the comparison with WOUDC archive. The difference between the WOUDC archives records processed in the NOAA system and WinDobson system are presented graphically in Fig. 9.

**Boulder, Colorado, USA (40°N, 105°W, NDACC station)**

Dobson observations were started at the University of Colorado east campus in 1966. Earlier observations were made either at the National Center for Atmospheric Research or at the Table mountain facility north of Boulder. The station was moved to the David Skaggs Research Center in 1999. Multiple instruments have been used here in the record, especially prior to the automation of Dobson instrument D061 in 1980. The observations made after 1980 automation do not include CC' zenith observations. The instrument was rebuilt with the WinDobson automation, but the data was processed in the NOAA system until the beginning of 2015. There is data in the WOUDC archive prior to 1966, but not connected to a calibration. The data for July 2013 to July 2014 are missing from the WOUDC and NDACC archives. The periods 1992-1996, and 1998-2005 were not processed or archived using the correct calibration information. The difference between the WOUDC and NDACC archives records processed in the NOAA system and WinDobson system are presented graphically in Fig. 10. The Instrument's calibration is tracked more closely than at other stations, as the World Standard Dobson D083 is kept in Boulder.

**Wallops Island Flight Center, Virginia, USA (38°N, 76°W, NDACC Station)**

Dobson observations were started at WIFC in 1967 as support for balloon and rocket borne experiments. The station has moved several times to different sites within the facility. Since 1995 only ADDS observations are

made to support ozonesonde flights.  There are periods in the WOUDC and NDACC archives with either missing data, or archived with incorrect calibration information applied. The difference between the WOUDC and NDACC archives records processed in the NOAA system and WinDobson system are presented graphically in Fig. 11.

## NOAA/ESRL/GMD Observatory, Barrow, Alaska, USA (71°N, 157°W)

Dobson observations at the NOAA observatory began in 1973.  The instrument was out of operation between 1983-1986 due to lack of funding.  The difference between the WOUDC archive processed in the NOAA system and WinDobson system are presented graphically in Fig. 12.  The station's weather is far cloudier than at other stations, with the station reporting 58% ZS observations.  The change in the method for retrieving TOC from these observations is evident in the variability in the differences between the archives.

## NOAA/ESRL/GMD Observatory, American Samoa (14°S, 171°W, NDACC Station)

Dobson observations were started at the NOAA observatory in 1976.  The station is in a warm, humid marine environment which caused instrument degradation in the early part of the record.  The original processing pre-1995 was not standard and not repeatable.  The periods where the N-tables were reconstructed from the results of the special processing in 1995 are indicated by the yearly N-table triangles.   An earthquake and tsunami on the 29 September 2009 damaged the station and instrument and observations were interrupted for several years.  The difference between the WOUDC and NDACC archives records processed in the NOAA system and WinDobson system are presented graphically in Fig. 13.   The period 1999-2001, were not processed or archived using the correct calibration information.   The WOUDC and NDACC were not completely updated after observations were restarted, due to perceived instrument problems which since have been resolved.

## Fresno and Hanford, California, USA (36°N, 120°W)

Dobson observations were started at the Fresno Weather Service Office, California, (37°N, 120°W) in 1982, with observations starting the next year.  The Weather Service Office was moved to Hanford in March of 1995.  The difference between the WOUDC and NDACC archives records processed in the NOAA system and WinDobson system are presented graphically in Fig. 14.  There are very few issues with the Fresno and Hanford record.

## Observatoire de Haute-Provence, France (44°N, 6°E, NDACC Station)

Dobson observations were started at the Observatoire de Haute-Provence (Station Géophysique Gérard Mégie) in 1983, with an automated instrument.  This instrument was updated to the WinDobson automation and data processing in 2014.  The station and instrument are operated by the Centre National de la Recherché Scientifique, CNRS.  The period of 1990 to 1999 was reprocessed to account for calibration drift, but has not yet been updated in WOUDC and NDACC. The difference between the WOUDC and NDACC archives records processed in the NOAA system and WinDobson system are presented graphically in Fig. 15.   The instrument was damaged several time in its history; inspection within WinDobson resulted in the removal of days from inclusion in the record.  This produced several months of higher differences (February 2013, for example.)

**Perth Airport, Western Australia, Australia (32°S, 116°E)**

Dobson observations were started originally in 1969 at Perth Airport weather radar, Perth Western Australia, then the NOAA automated instrument D081 was installed in 1984. The instrument is operated by the Australian Bureau of Meteorology (BoM). In the late 1990s, the station was moved to the newly constructed Weather Station. There are periods of missing data in the WOUDC archive. The period after 2012 in the WOUDC archive does not have correct calibration information, as the BoM recalibrated the instrument, and this information was not included in NOAA's database of calibrations. The difference between the WOUDC and NDACC archives records processed in the NOAA system and WinDobson system are presented graphically in Fig. 16.

**Lauder, Central Otago, New Zealand (45°S, 170°E, NDACC Station)**

Dobson observations began in early 1987 at the Research station in Central Otago, South Island, New Zealand. The instrument is operated by New Zealand's National Institute of Water and Atmospheric Research (NIWA). The station's time zone is UTC + 12, which means the UTC day changes at Local Standard Time 12 noon. The calculation of the local day and UTC day for reporting the selected value was incorrect prior to 1992, indicated by the higher scatter in the comparison of the old and new archives in that time period. Also, a selected value could be from the afternoon of one local day, and the representative value from the morning of the following local day while still being in the same UTC day. When inspected during the WinDobson processing, the instrument record between 2006 through 2011 revealed rain damage following reinstallation shortly after the 2006 intercomparison in Melbourne. The 2012 calibration information determined before the rebuilding of the instrument was used to process the data in WinDobson during 2006 through 2012. The inspection also determined that the calibration was stable from 1992 to 2006, while the instrument calibration was checked in 1997 and 2001. The WOUDC and NDACC records are not yet updated. The instrument was rebuilt at the beginning of 2012, and has been operated with the data reduction in WinDobson since that time. The difference between the WOUDC and NDACC archives records processed in the NOAA system and WinDobson system are presented graphically in Fig. 17.

**Conclusions**

NOAA has submitted nearly a half century's data into the WOUDC and NDACC archives. Personnel and data processing protocols changed many times throughout that period, and knowledge of early techniques was slowly being lost. Furthermore NOAA personnel tended to use a larger and more comprehensive data base when performing research, so the accuracy of data within the WOUDC and NDACC archives were seldom questioned. Our experiences in the investigation of the long-term archived NOAA Dobson data records should alert other Dobson data producers of the importance of regular review and intercomparisons of the archived station's records residing at multiple archives. The Dobson station data processing procedures and software tends to change over time after the new knowledge or technology becomes available. Although the reprocessing of historical datasets is extremely difficult due to lost documentation or even raw data, the benefit of the investigation is record's homogenization and adjustment to conform to the WMO/GAW operating procedure guidance (Komhyr and Evans, 2008). With the advent of Windobson software and its newer technique for calculating TOC from zenith observations and selecting representative observations, we felt it was prudent to reprocess all previous

measurements for the sake of homogeneity. It also seemed logical to compare and replace data within the WOUDC and NDACC archives with the newly reprocessed data. The overall changes are small (~0.1% offset), but several individual stations have a larger offset (Maximum 0.7%) driven by the changes in the ZC reduction polynomials. During comparisons between WinDobson dataset and the existing NDACC and WOUDC archives, we were able identify periods with either missing data or incorrectly processed data. The differences between the historical and the new version of Dobson data have overall small offsets and trends (Table 2), but within the long-term record there are periods with greater differences of which researchers should be aware (see Fig.s 4 through 16, and description of the individual station histories.). The paper includes a section that describes individual station histories, which provides information on specific-to-station updates and their effects on the total ozone record. The offsets and trends for differences between the old and the new version of the data are not the same for WOUDC and NDACC archives, as the NDACC set of data is not a perfect match to the one available from the WOUDC archive . For example, Wallops Island NDACC record is 1995-2014, while the WOUDC record is 1967-2014. When the NDACC and WOUDC archives are updated, these archived datasets will be complete and homogenized. Moreover, after all calibrations and the applicable periods were reviewed, the history method of applying calibrations to all of the instruments in the networks has been standardized. The complete (all observations) new WinDobson operational database, available to researchers on request, will allow investigators to improve the accuracy and consistency of the Dobson retrieval algorithms and stations records. The expectation is that in the future the WOUDC and NDACC archives will consist of all observation results.

**Acknowledgements.**

The authors would like to acknowledge the work done in the past by such people as Walter Komhyr, Robert Grass, Kent Leonard and others in establishing the US network.

The Dobson observations at Lauder are supported through NIWA's core research funded by the NZ Ministry of Business, Innovation and Employment; at Perth by the Bureau of Meteorology, an Executive agency of the Australian Government; and at l'Observatoire du Haute Provence by the National Center for Scientific Research (CNRS), under the responsibility of the French Ministry of Education and Research. Support for the updating of the automation at several NDACC sites was provided by the NOAA Joint Polar Satellite System (JPSS) Calibration/Validation program.

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

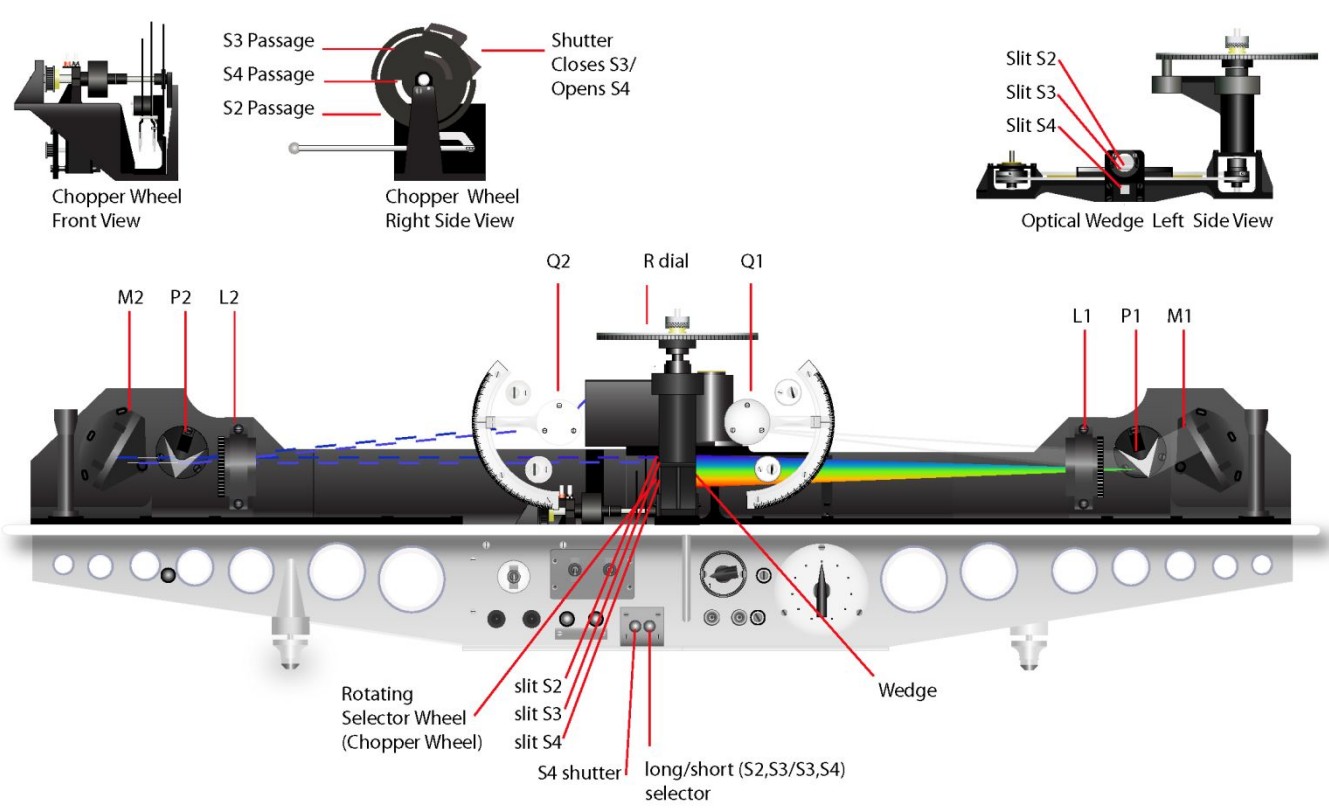

**Figure 1**: Diagram of Dobson Instrument, with cover omitted from view (some components shown are actually mounted in the cover.)

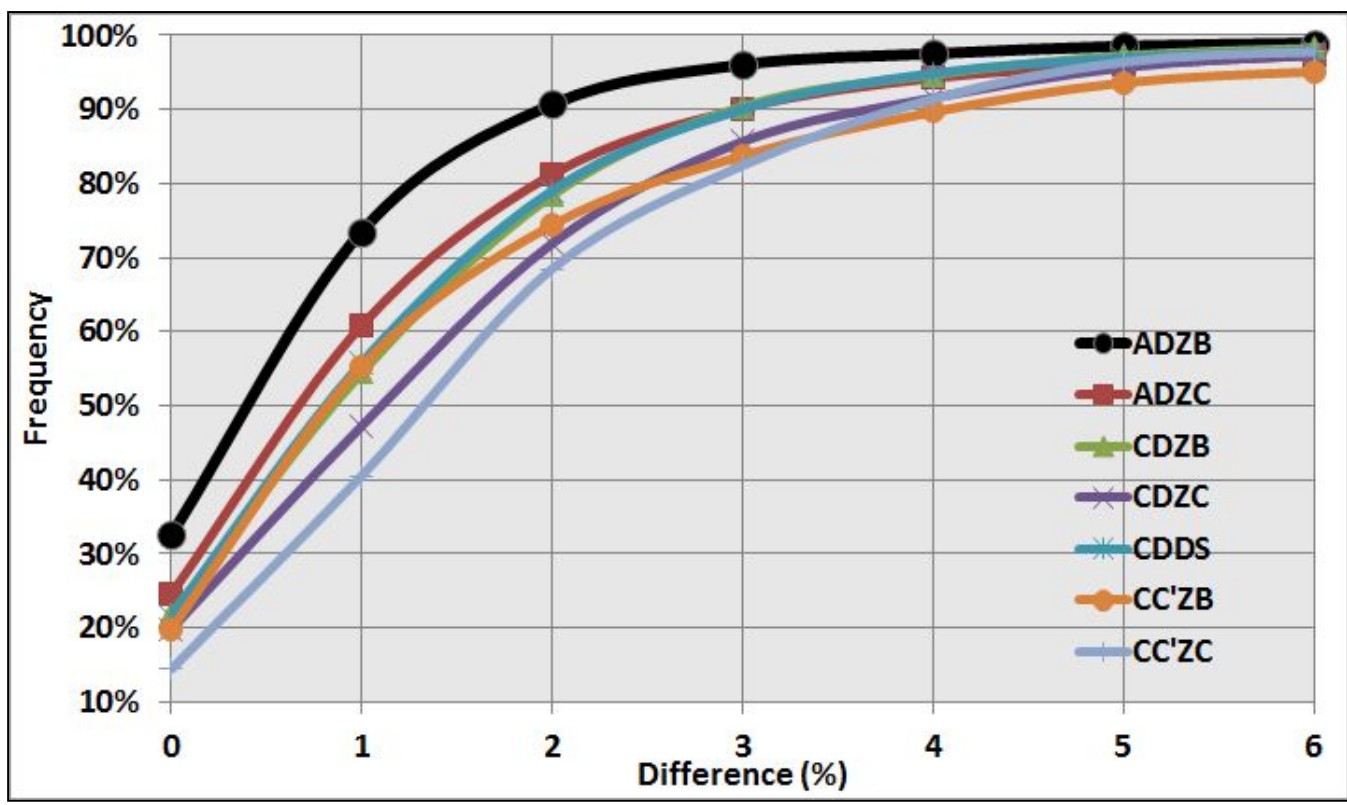

**Figure 2.** Distribution of cumulative differences between results from direct sun (ADDS) compared to zenith measurements on the same day. The frequency of compared zenith and ADDS total ozone (y-axes) is accumulated between 0 to 6 % (X-axes). Results are shown for other types of zenith sky measurements denoted by colors in the legend. Results are the average of 12 stations in the US network except for the CC' results, which is based on the SPO data record.

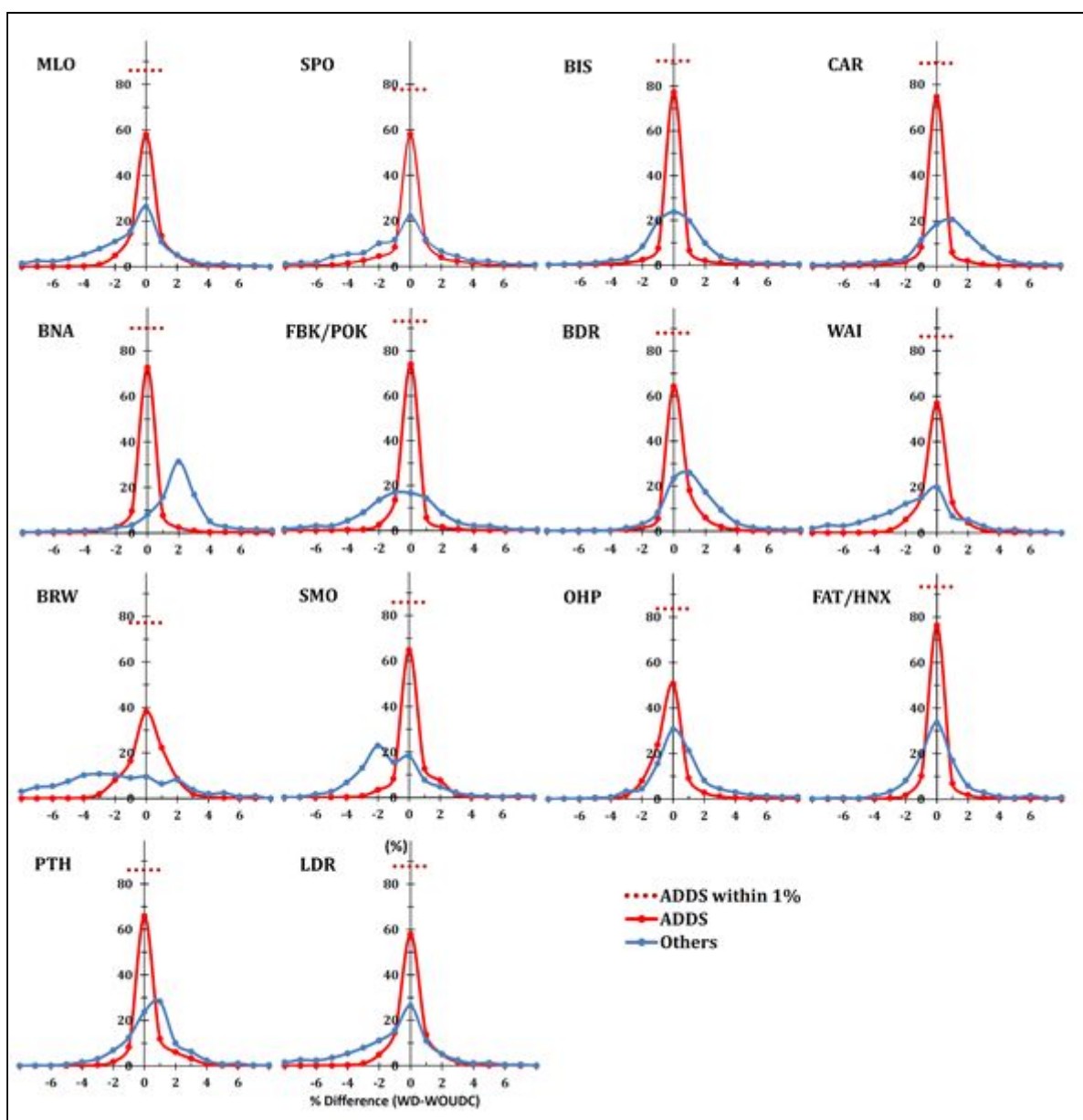

**Figure 3.** The Probability of a daily value changing by a particular percentage for each station. The red line is for ADDS type observations; the blue for all other types**;** the horizontal line is the percent of ADDS in the range **±1%** differences.

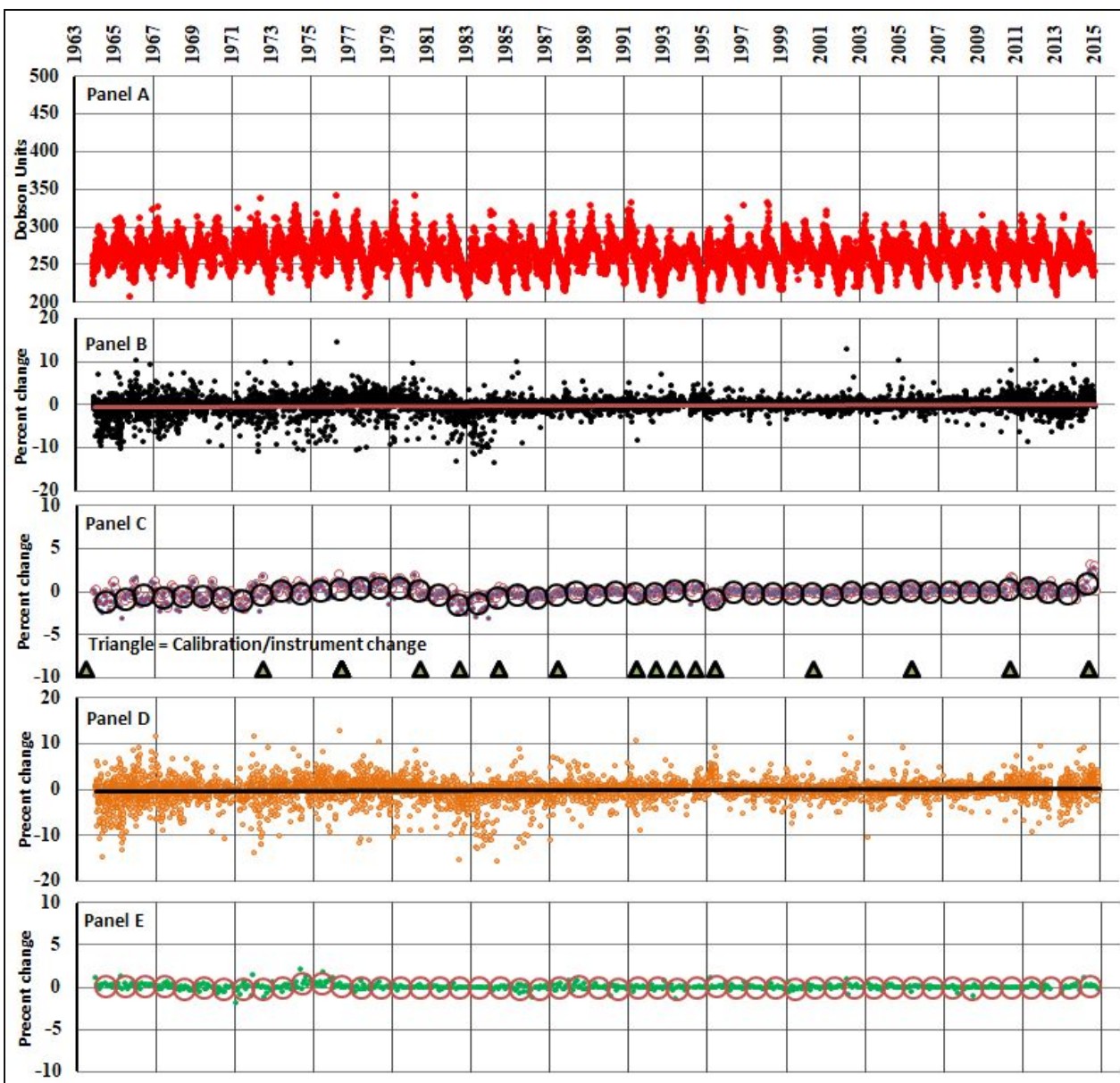

Figure 4: Graphic representation of the changes in the Mauna Loa Observatory, Hawai'i, USA (19°N, 156°W, NDACC Station) record after the conversion into WinDobson processing. First panel: The time record of total ozone measured at the station from the start of observations through to 2015 (or until station was converted to WinDobson processing. Second Panel: percent difference between daily WinDobson total ozone records compared to the WOUDC record, (WinDobson-WOUDC). The red line is a linear fit. Third panel: the same as the second but for monthly and yearly averages (based on all the values in the month in each data set). The small white circles are averages made from DS observations only; the red symbols represent averages using all Dobson

total ozone records; the large black open circles are yearly averages of all observations, based on monthly averages. Large triangle symbols indicate major calibration or instrument changes that lead to creating the new N-tables; however, not all calibrations checks of the station record are shown. For NDACC Stations only: The fourth panel is the same as the second panel but for comparisons with the data archived at NDACC center (WinDobson-NDACC). The black line is a linear fit. The fifth panel is the same as the third panel for comparisons with the NDACC archived monthly and yearly averages. NDACC values are not recorded as observation type.

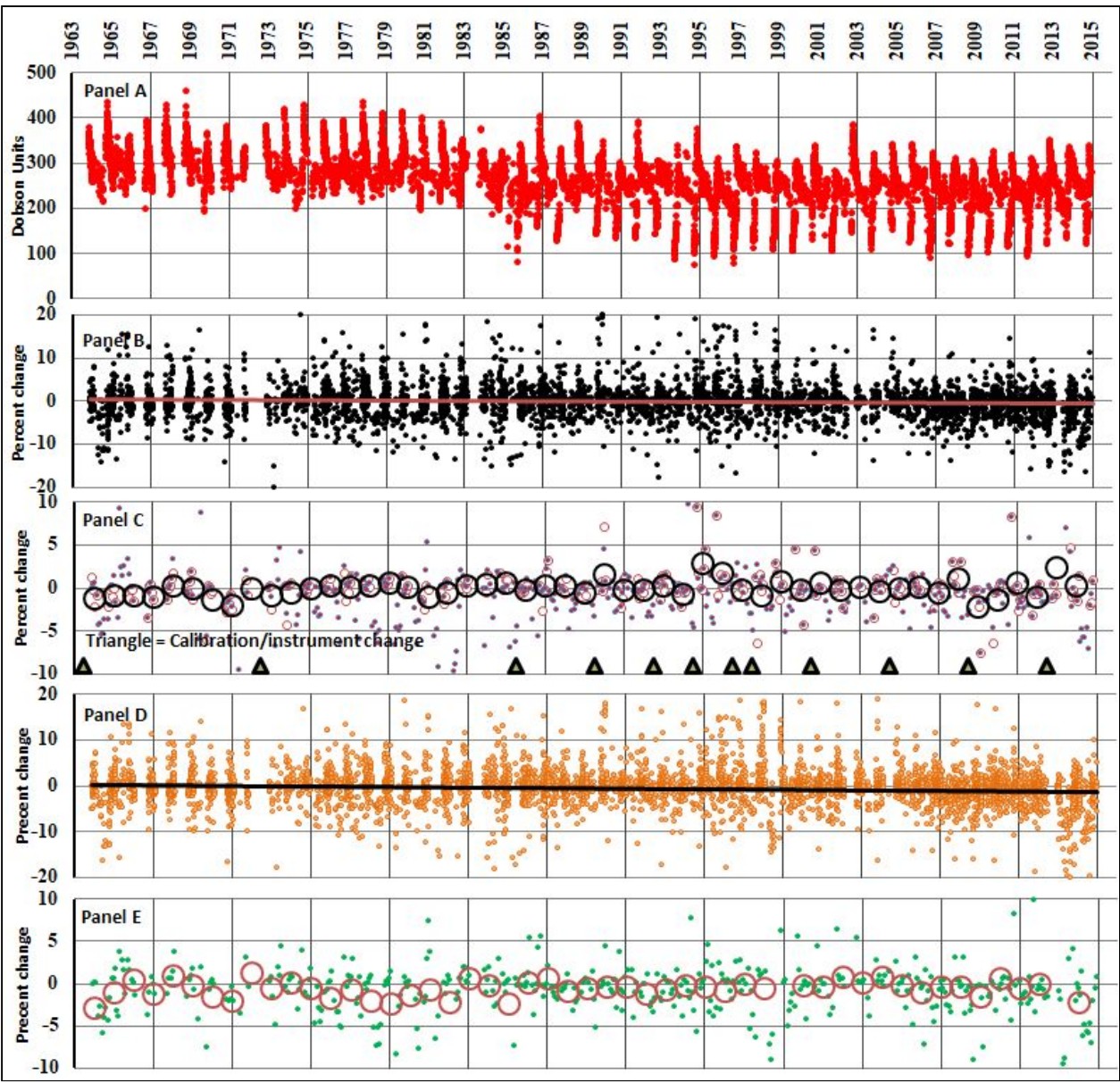

Figure 5: Graphic representation of the changes in the South Pole, Antarctica (90°S, 59°E, NDACC Station) record with the conversion into WinDobson processing.

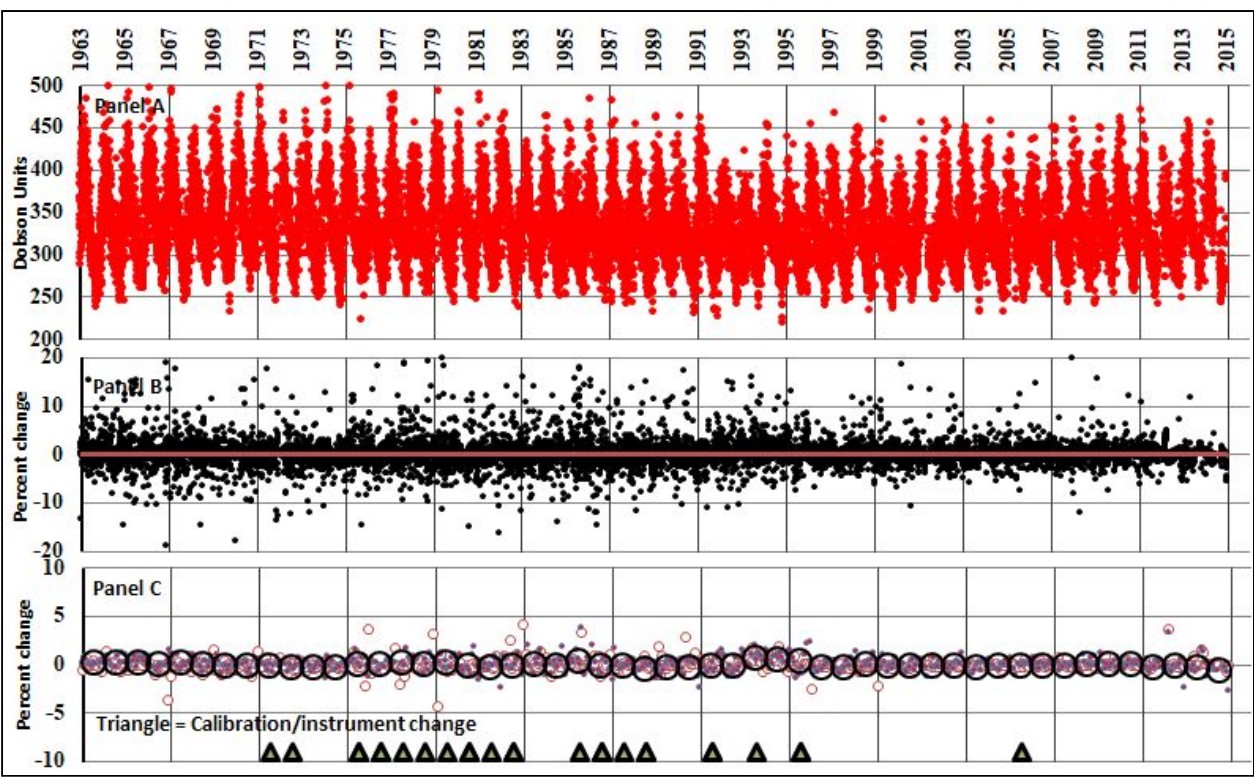

Figure 6: Graphic representation of the changes in the Bismarck, North Dakota, USA (47°N, 101°W) record with the conversion into WinDobson processing.

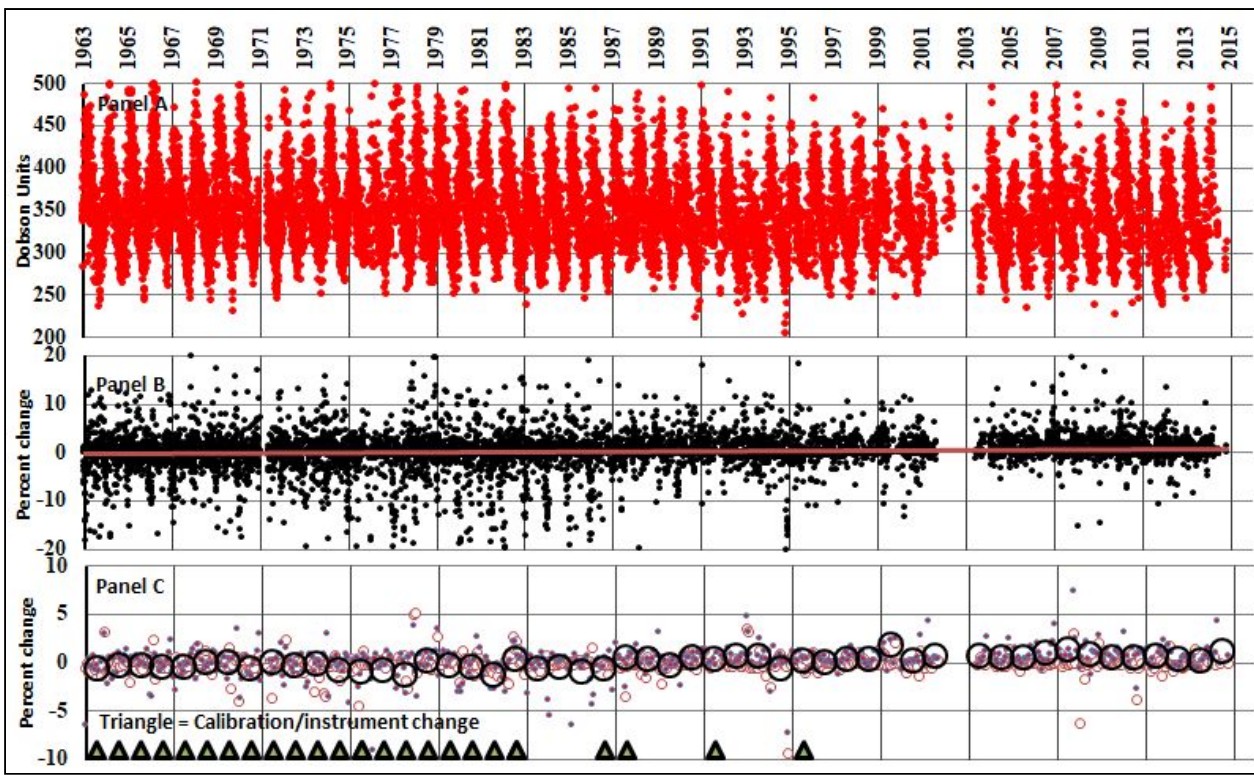

Figure 7: Graphic representation of the changes in the Caribou, Maine, USA (47°N, 68°W) record with the conversion into WinDobson processing.

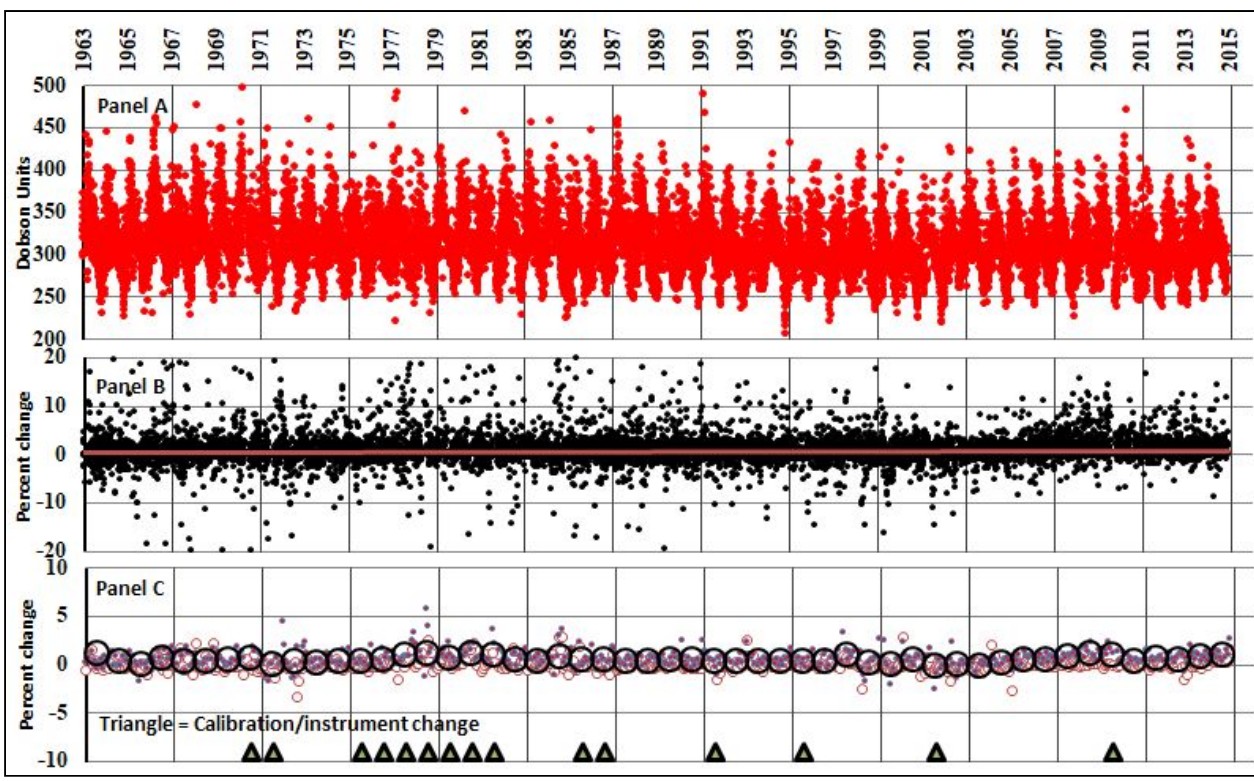

Figure 8: Graphic representation of the changes in the Nashville, Tennessee, USA (36°N, 87°W) record with the conversion into WinDobson processing.

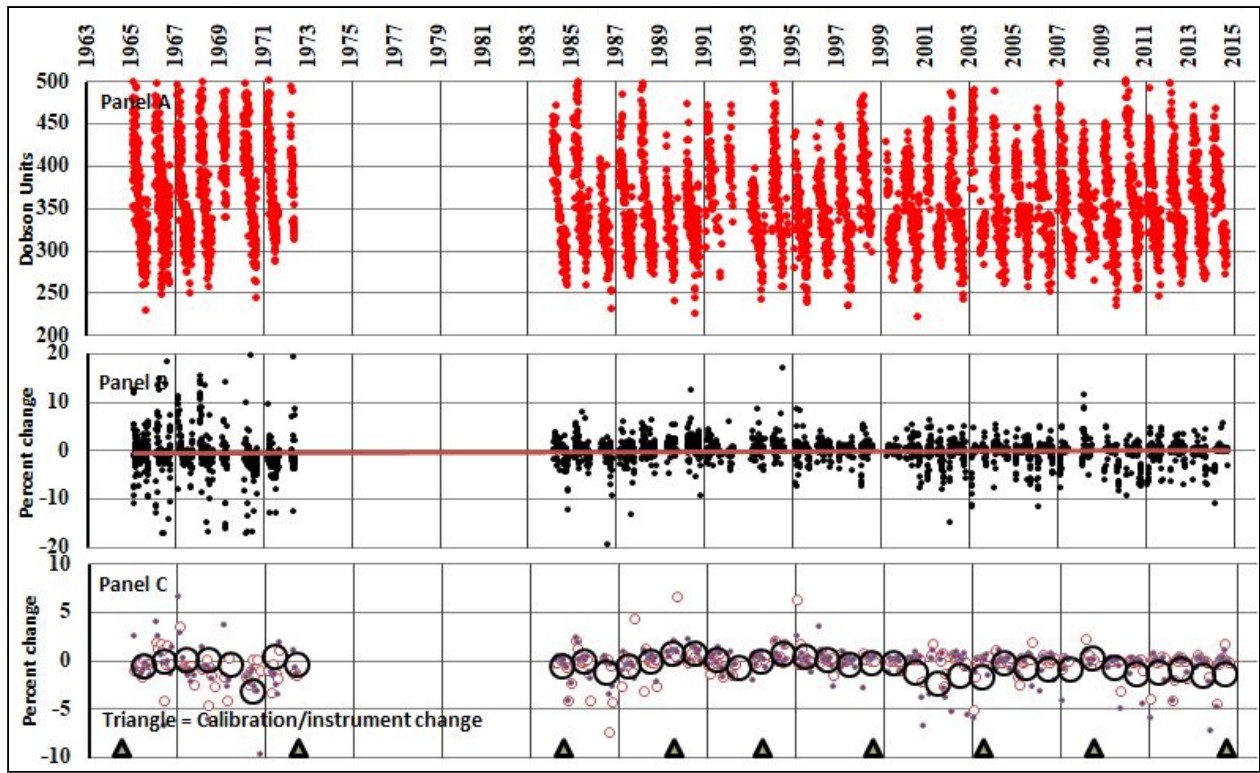

Figure 9: Graphic representation of the changes in the Fairbanks, Alaska, USA (65°N, 148°W) record with the conversion into WinDobson processing.

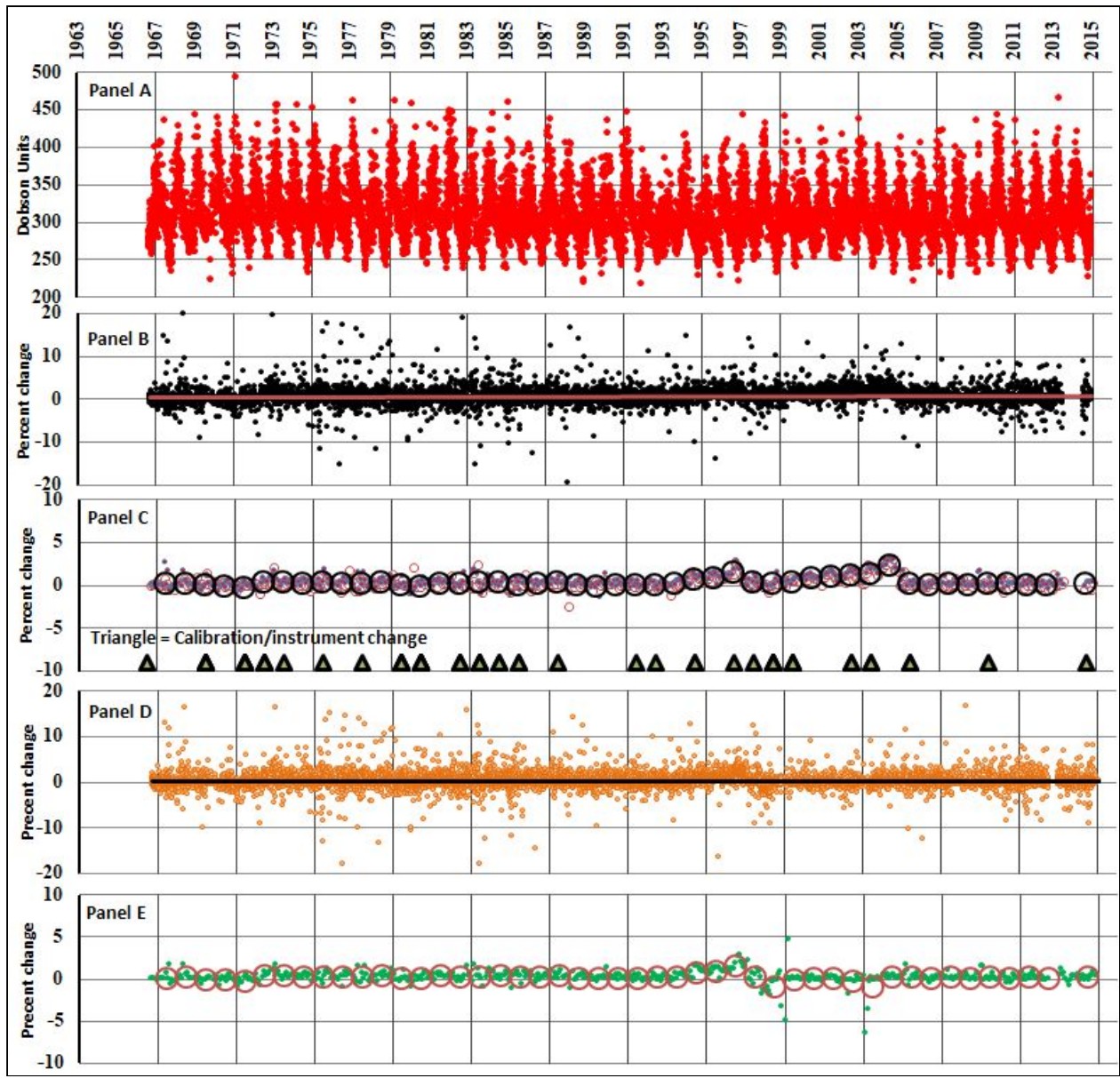

Figure 10: Graphic representation of the changes in the Boulder, Colorado, USA (40°N, 105°W, NDACC station) record with the conversion into WinDobson processing.

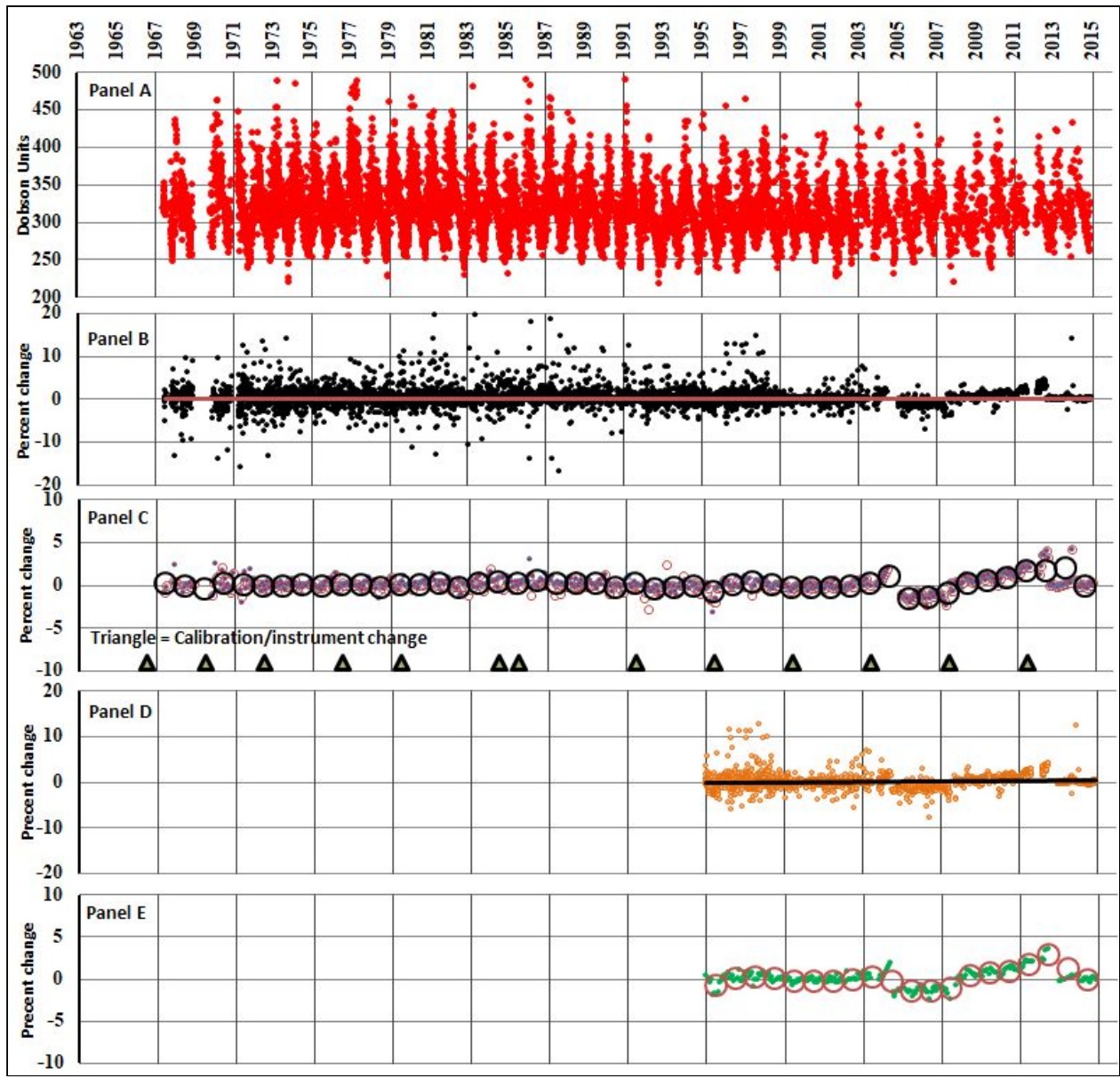

Figure 11: Graphic representation of the changes in the Wallops Island Flight Center, Virginia, USA (38°N, 76°W, NDACC Station) record with the conversion into WinDobson processing.

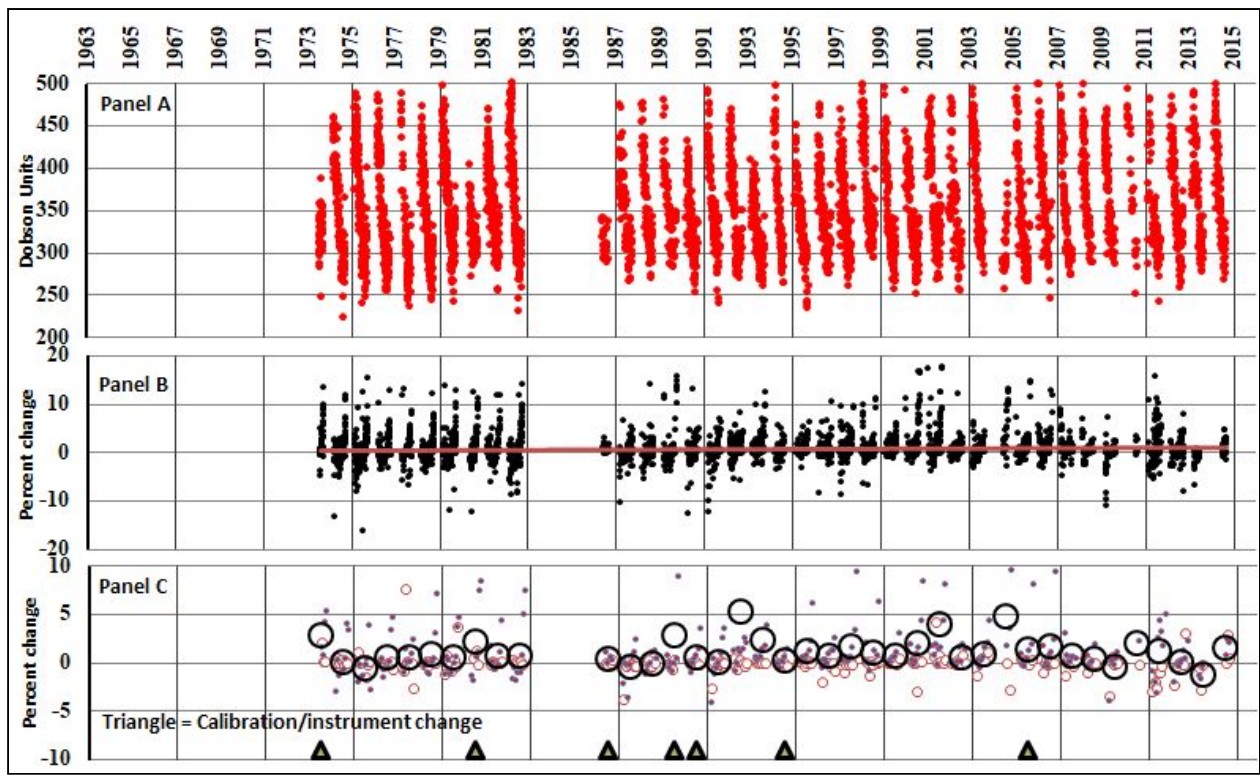

Figure 12: Graphic representation of the changes in the NOAA/ESRL/GMD Observatory, Barrow, Alaska, USA (71°N, 157°W) record with the conversion into WinDobson processing.

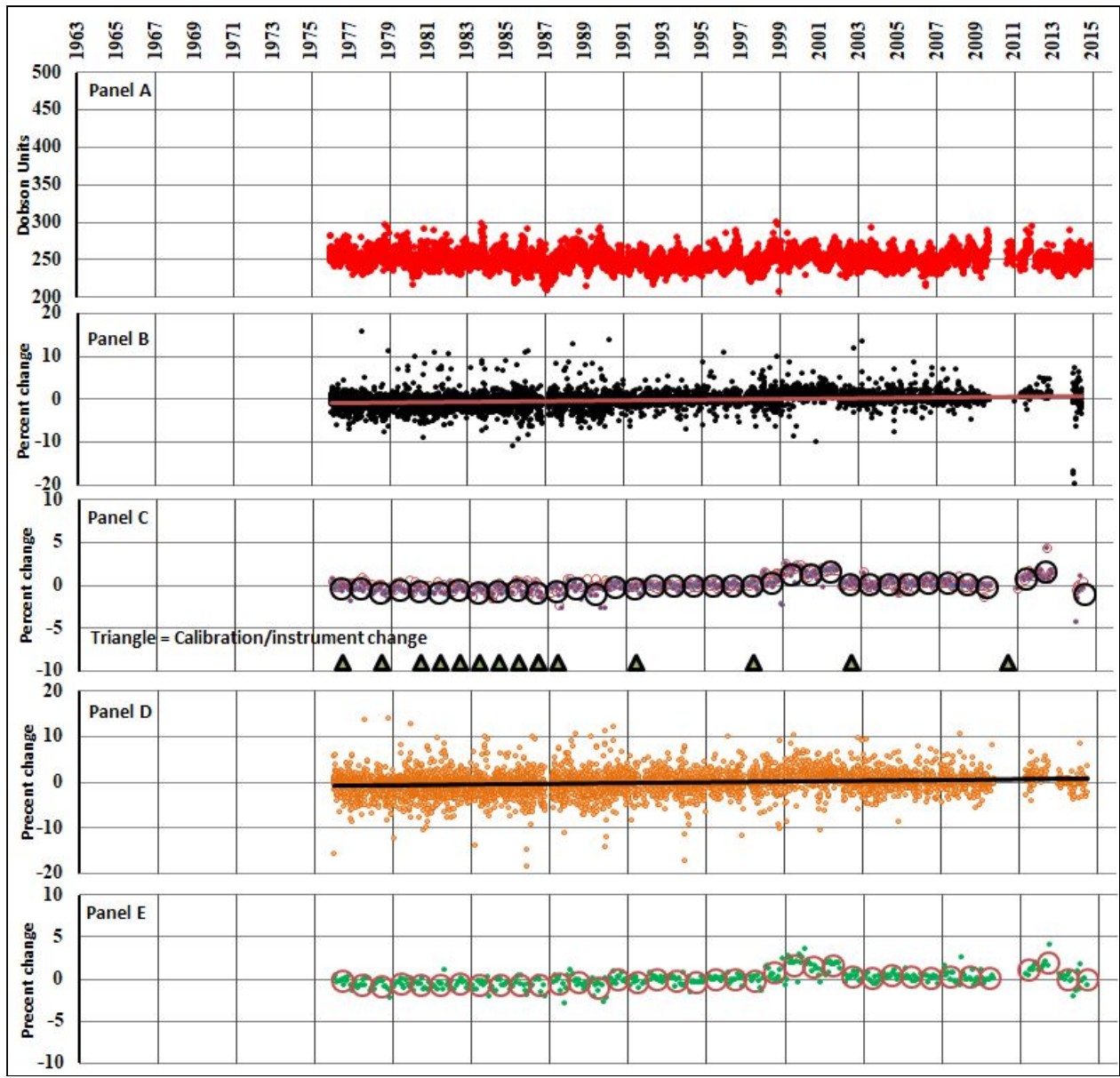

Figure 13: Graphic representation of the changes in the NOAA/ESRL/GMD Observatory, American Samoa (14°S, 171°W, NDACC Station) record with the conversion into WinDobson processing.

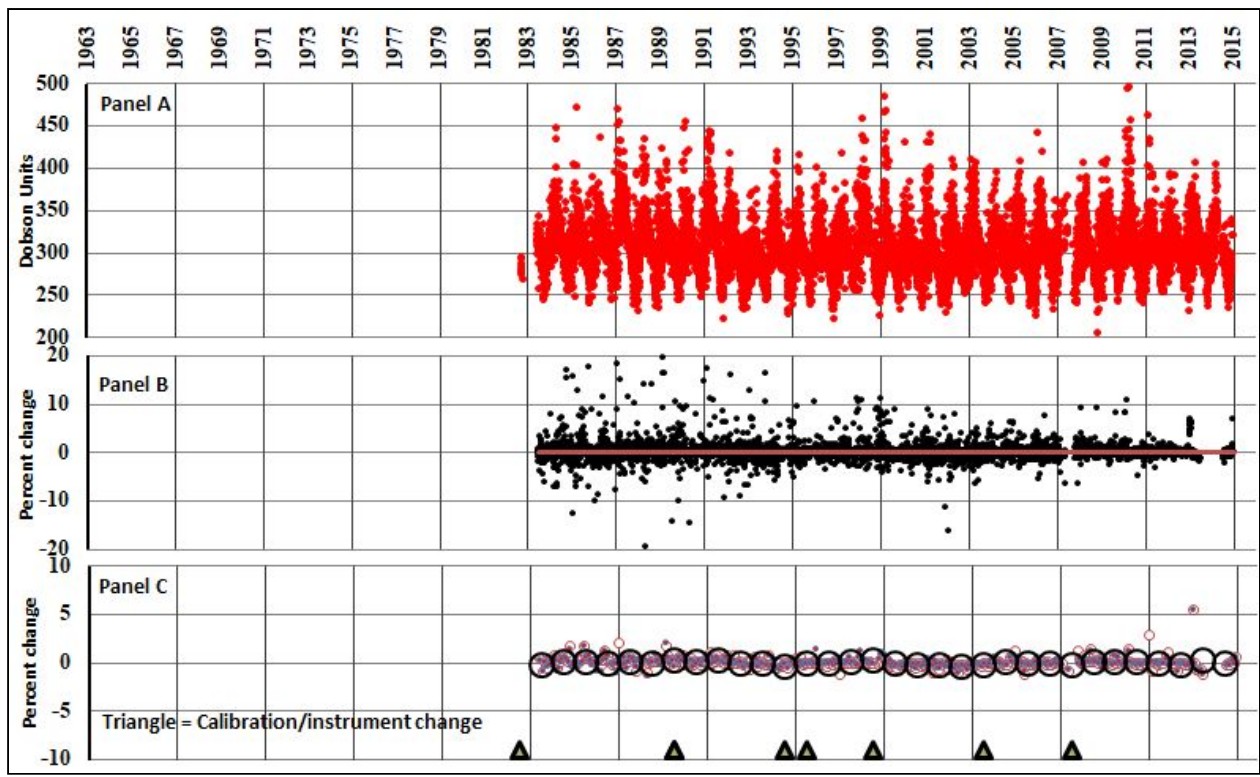

Figure 14: Graphic representation of the changes in the Fresno and Hanford, California, USA (36°N, 120°W) record with the conversion into WinDobson processing.

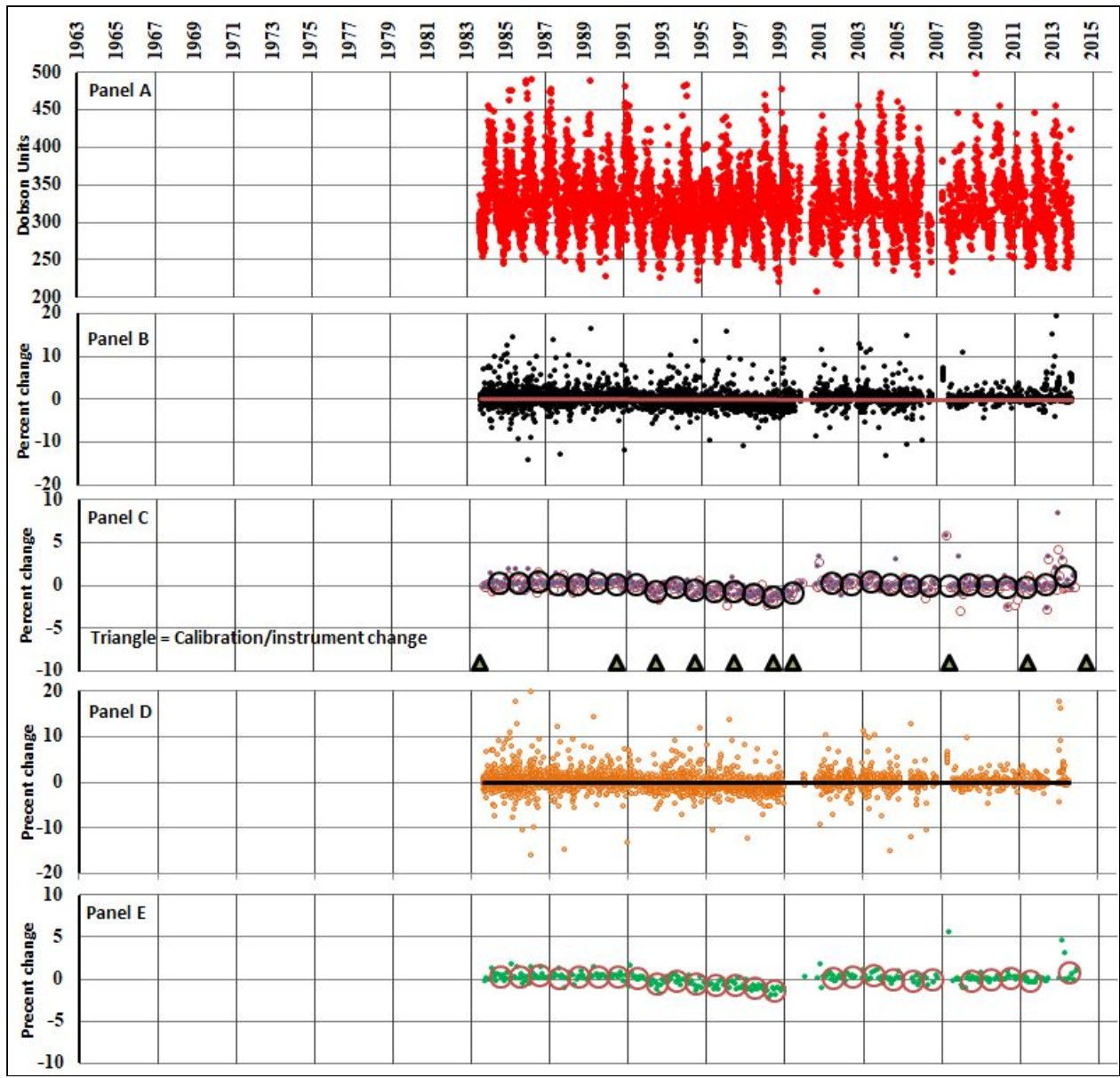

Figure 15: Graphic representation of the changes in the Observatoire de Haute-Provence, France (44°N, 6°E, NDACC Station)record with the conversion into WinDobson processing.

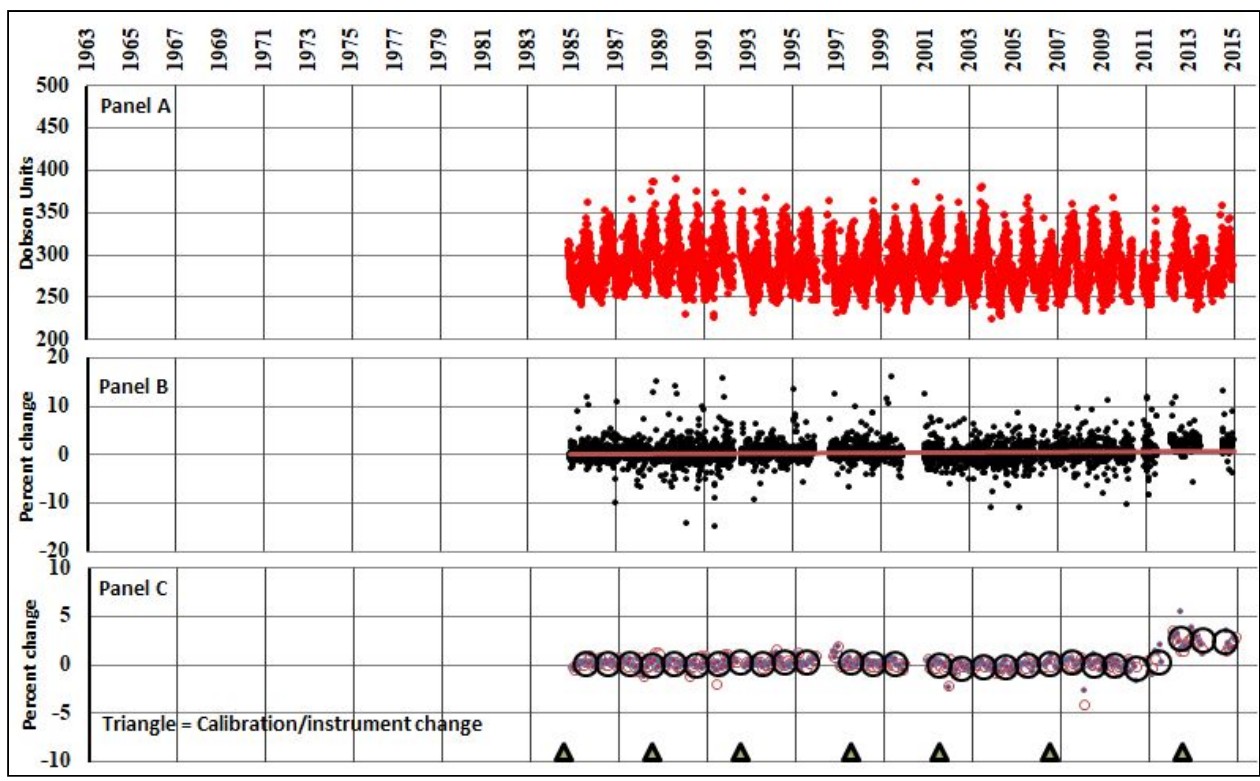

Figure 16: Graphic representation of the changes in the Perth Airport, Western Australia, Australia (32°S, 116°E) record with the conversion into WinDobson processing.

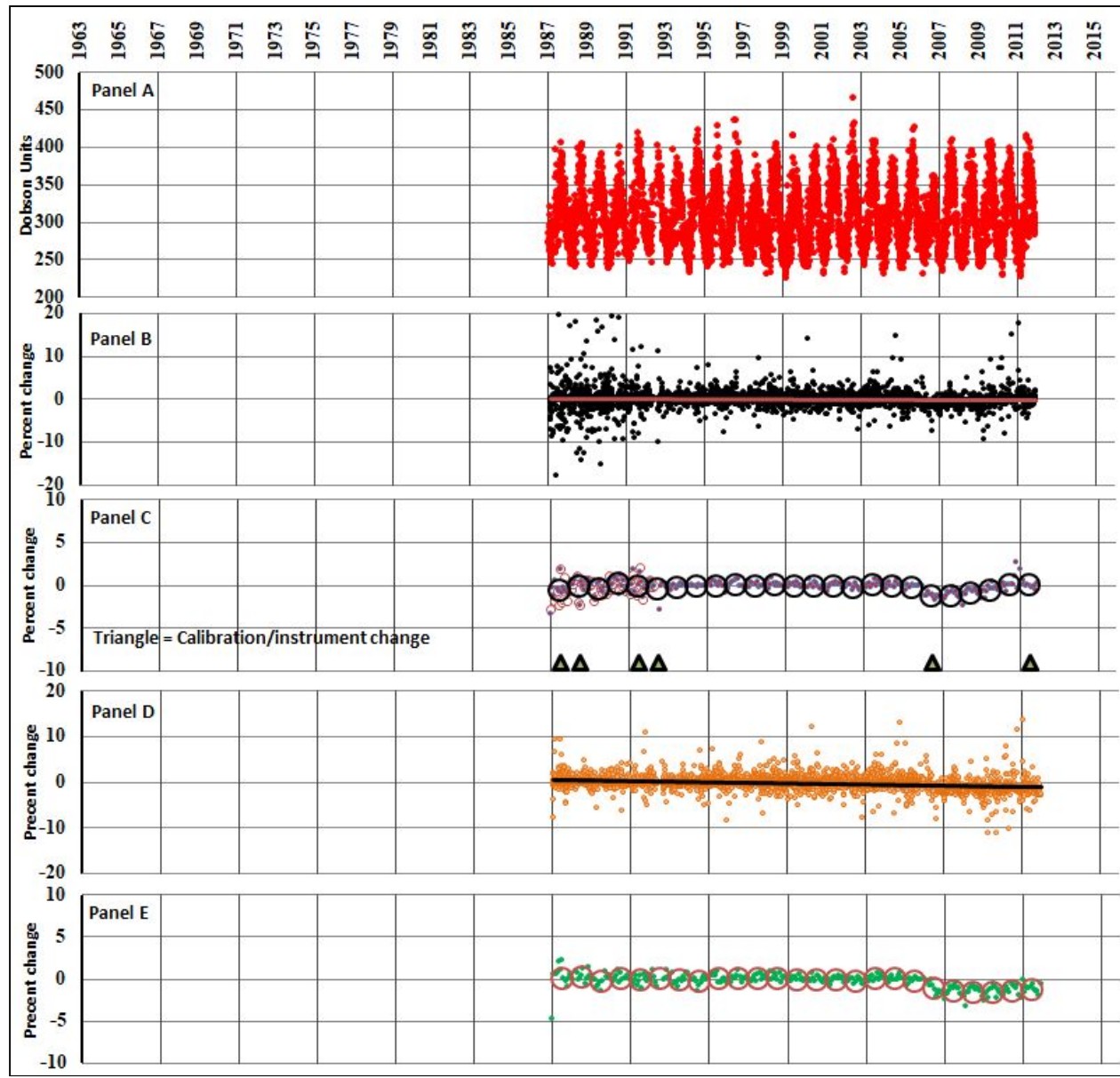

Figure 17: Graphic representation of the changes in the Lauder, Central Otago, New Zealand (45°S, 170°E, NDACC Station) record with the conversion into WinDobson processing.

processing.

| Station Name | NOAA Station Code and WMO station Number | Station Dobson Record Started | Responsible Organizations (Archives) | Current Automation Status |
|---|---|---|---|---|
| Mauna Loa GMD Observatory | MLO 31 | 1963 | NOAA (NDACC and WOUDC) | WinDobson Full |
| South Pole | SPO 111 | 1963 | NOAA (NDACC and WOUDC) | NOAA Semi-Auto |
| Bismarck, North Dakota | BIS 19 | 1962 | NOAA | NOAA Semi-Auto |
| Caribou, Maine | CAR 20 | 1962 | NOAA | NOAA Semi-Auto |
| Nashville, Tennessee | BNA 106 | 1962 | NOAA | NOAA Semi-Auto |
| Fairbanks, Alaska | FBK; POK 105; 217 | 1965 | NOAA; University of Alaska | WinDobson Full |
| Boulder, Colorado | BDR 67 | 1966 | NOAA (NDACC and WOUDC) | WinDobson Full |
| Wallops Is., Virginia | WAI 107 | 1967 | NOAA; NASA (NDACC and WOUDC) | NOAA Semi-Auto |
| Barrow GMD Observatory | BRW 199 | 1973 | NOAA | NOAA Semi-Auto |
| American Samoa, GMD Observatory | SMO 191 | 1976 | NOAA (NDACC and WOUDC) | NOAA Semi-Auto |
| Haute Provence, France | OHP 40 | 1983 | NOAA; Centre National de la Recherche Scientifique, (NDACC and WOUDC) | WinDobson Full |
| Fresno and Hanford, California | FAT; HNX 244; 341 | 1983 | NOAA | NOAA Semi-Auto |
| Perth, Australia | PTH 159 | 1984 | NOAA; Australian Bureau Meteorology | NOAA Full |
| Lauder, New Zealand | LDR 256 | 1987 | NOAA; National Institute for Water and Atmosphere | WinDobson Full |

| | | | (NDACC and WOUDC) | |

Table 1. Current Stations in the NOAA Network using Dobson Ozone Spectrophotometers

| Station Code | Offset WinDobson-WOUDC | Linear Trend WinDobson- WOUDC Per Year | Offset WinDobson-NDACC | Linear Trend WinDobson- NDACC Per year |
|---|---|---|---|---|
| MLO | -0.1% ± 1.6% | +0.014 ± 0.001% | -0.1% ± 1.8% | +0.015 ± 0.001% |
| SPO | -0.0% ± 4.0% | -0.016 ± 0.003% | -0.5% ± 6.9% | -0.026 ± 0.006% |
| BIS | +0.1% ± 2.2% | -0.004 ± 0.001% | N/A | N/A |
| CAR | +0.2% ± 3.2% | +0.022 ± 0.002% | N/A | N/A |
| BNA | +0.6% ± 2.7% | +0.002 ± 0.001% | N/A | N/A |
| FBK | -0.4% ± 2.8% | +0.033 ± 0.003% | N/A | N/A |
| BDR | +0.3% ± 1.7% | +0.007 ± 0.001% | +0.3% ± 1.5% | -0.001 ± 0.001% |
| WAI | -0.1% ± 3.3% | +0.024 ± 0.003% | +0.0% ± 1.6% | +0.032 ± 0.006% |
| BRW | +0.7% ± 2.8% | +0.011 ± 0.004% | N/A | N/A |
| SMO | -0.1% ± 1.7% | +0.042 ± 0.002% | -0.1% ±2.3% | +0.042 ± 0.002% |
| OHP | -0.1% ± 1.8% | -0.002 ± 0.003% | -0.1% ± 1.7% | -0.004 ± 0.003% |
| FAT/HNX | +0.0% ± 1.5% | -0.003 ± 0.002% | N/A | N/A |

| | | | | |
|---|---|---|---|---|
| PTH | +0.3% ± 1.6% | +0.022 ± 0.002% | N/A | N/A |
| LDR | -0.1% ± 1.8% | -0.022 ± 0.003% | -0.3% ± 1.3% | -0.066 ± 0.002% |

**Table 2.** Statistics of the overall differences between WOUDC and NDACC records and WinDobson record (WinDobson-WOUDC, NDACC).

| % Difference from ADDS | ADZB (%) | ADZC (%) | CDZB (%) | CDZC (%) | CDDS (%) | CC'ZB (%) | CC'ZC (%) |
|---|---|---|---|---|---|---|---|
| 0 | 33 | 25 | 22 | 20 | 22 | 20 | 14 |
| 1 | 74 | 61 | 54 | 47 | 56 | 55 | 41 |
| 2 | 91 | 81 | 78 | 72 | 79 | 74 | 68 |
| 3 | 96 | 90 | 90 | 86 | 90 | 84 | 82 |
| 4 | 98 | 94 | 95 | 92 | 95 | 90 | 91 |
| 5 | 99 | 96 | 97 | 96 | 97 | 94 | 96 |
| 6 | 99 | 97 | 98 | 97 | 98 | 95 | 98 |
| 7 | 99 | 98 | 99 | 98 | 98 | 96 | 98 |
| 8 | 100 | 99 | 100 | 98 | 99 | 96 | 98 |
| Frequency | ADZB | ADZC | CDZB | CDZC | CDDS | CC'ZB | CC'ZC |
| 85 | 1.5 | 2.3 | 2.5 | 3.0 | 2.5 | 3.3 | 3.3 |
| 90 | 1.9 | 3.0 | 3.0 | 3.6 | 3.0 | 4.0 | 3.8 |

**Table 3.** Displayed is the cumulative agreement in percent for specific ZS and CDDS results compared to ADDS results on the same day. For example, for an agreement of 2% occurs in 91% of the cases for ADZB observations. Displayed are the average of 12 stations in the NOAA network (Barrow, Fairbanks, Caribou, Bismarck, Haute Provence, Boulder, Wallops Island, Mauna Loa, Tutuila, Perth, Lauder and South Pole).