# Peer review of "The US Dobson Station Network Data Record Prior to 2015, Re-evaluation of NDACC and WOUDC archived records with WinDobson processing software"

_Atmospheric Chemistry and Physics, 2017_

## Author Comment (AC1) · 10 May 2017

The following paragraph was mistakenly omitted in the production manuscript under the Acknowledgements heading:

The Dobson observations at Lauder are supported through NIWA's core research funded by the NZ Ministry of Business, Innovation and Employment; at Perth by the Bureau of Meteorology, an Executive agency of the Australian Government; and at l'Observatoire du Haute Provence by the National Center for Scientific Research (CNRS), under the responsibility of the French Ministry of Education and Research.

Support for the updating of the automation at several NDACC sites was provided by the NOAA Joint Polar Satellite System (JPSS) Calibration/Validation program.

---

## Referee Comment (RC1) · Anonymous Referee #1 · 24 May 2017

This reviewer has several issues with this paper but all are easily remedied. The primary issue is that the data are not publically available so the reviewer(s) are not able to check any claims made by the authors. The policies of ACPD are a bit vague but many publications will not allow submission of papers citing proprietary data. I suggest putting the new data somewhere where it can be accessed before the release of this paper.

The second issue is concerning the conclusion section. This section is extremely underwhelming. The reader really wants to know WHY this work was done and to how

the Dobson data has changed in a scientific sense. There are large differences in the data at high latitudes (+/- 10 percent!) which should be very easily studied. A couple of paragraphs statistically comparing the new & old datasets to the satellite overpasses for those Dobson stations would be highly useful as would a plot or two and that added work would make this paper much stronger and more complete. Can you quantify how much better the new data are? Less noise? Fewer step functions? Less bias? This is the payoff for all that hard work analyzing and revising the old data.

Issue 3: When a new calibration or instrument repair was done how were the new calibration values applied? Were they put in as a step function or gradually introduced over time (linearly?) between known calibrations/changes?? Please explain.

Specific little changes recommended: Line 16: remove "for possible changes" Line 25: remove comma I recommend putting lines 36-8 after line 59 Line 57 & 80: you may want to define what the optical wedge is. Figure 3 is referenced before Figure 2. Please fix Line 149 is not a sentence Line 157 selected value of what? Line 182&235: Change was to were Line 207: And 'the' before Bismarck Line 236: use "There are data in the archive prior to 1966 but are not connected. . .. Line 237: change is to are Line 242: remove commas Line 247: remove comma Line 293: "station" appears twice.

All plots are daily averaged Dobson values or individual measurements? Please put units on plots!

---

## Referee Comment (RC2) · Anonymous Referee #2 · 7 Jun 2017

General Comments:

This technical note presents the re-evaluation of the total ozone data record derived by Dobson spectroradiometers operating by NOAA. The reprocessed data are compared with the data already deposited to the databases of WOUDC and NDACC. The manuscript includes important information from the history of the different stations and the problems encountered during their long term operation. I think it is a good practice to publish such information which is usually accessible only from the stations' personnel.

[Figure]

I see two main weaknesses in this paper. First, the WinDobson software package is not described adequately so that the reader cannot assess the differences in processing of the data compared to the traditional methods. The link provided as reference (end of page 3) does not help because it is a very brief slide presentation. Please, either provide a more suitable reference where the methods are described in detail, or include more details in the text.

Second, as the authors have long experience and deep knowledge of the Dobson retrieval algorithms and the instrument details, they tend to present their thoughts very briefly assuming that the reader has the same level of knowledge. I suggest to provide some more details so that even people who are not involved directly in the Dobson measurements can follow the paper easily.

Finally there are some rather minor presentation problems that are mentioned explicitly in my specific comments.

I suggest to accept this technical note for publication in ACP after taking into account the following suggestions in the revised manuscript:

Specific comments by line number:

2: Please change to ". . . intensity of solar radiation between . . ."

32-34: Something is missing in this sentence. The importance of the Dobson could not be demonstrated by "using Dobson units".

38: This handbook should be included in the References section.

41: Please insert here a reference as the meaning of the R and N values is understandable only from scientists experienced in the Dobson.

43: Would "applicability" be a better choice than "usefulness"?

73: Although traditionally ozone was archived as single daily values, nowadays individual values within a day are also available. Please revise this sentence accordingly.

[Figure]

73: "In this publication". Please clarify which publication you are referring to.

82: "a wide range". Please specify what "range" refers to.

93-94: Please provide a reference for the "statistical method" and "the set of rules" or include a description in the text.

94: Please specify to what quantity the "representative value" refers; is it "total ozone"?

106: What means "correct any differences"? Which data set was corrected; the old or the new?

118: Is this "multiplying factor" solar zenith angle dependent? Is it possible to provide an estimate of the uncertainty in the CD data after the adjustment?

130-158: The listed reasons for discrepancies between the old and the revised datasets are not always very clear. For example: - it is not clear how the mu-dependency affects the comparison. - is it true that drifts in the wedge calibration were not taken into account in the new dataset although they were considered in the original data? Wouldn't that mean that the new data are less accurate compared to the original?

159: Multiple archives of geophysical data is a concern because in many cases it is unknown if these datasets are the same. Therefore, it would be very useful to discuss differences between the WOUDC and NDACC archives at individual stations and give an indication of how important these differences are. This could be presented, for example, in the form of a probability plot (like those of Figure 2) or just as a number e.g., the percentage of differences smaller than 1%.

171: Table 2 appears after table 3. Please fix the numbering.

174: Figure 2 appears after Figure 3. Please fix the numbering.

186: Reference (Langley, 1984) is missing from the References list.

289: The conclusions section is very small. I suggest including a brief discussion referring to statistics of Table 2. For example it is mentioned that the trend is very small, but you could at least include the range, or mention in which stations the new dataset would affect the trend.

326: Figures 4 - 17: Please add labels and units in the vertical axes. It would be helpful to draw a light horizontal line at zero in the panels showing differences.

---

## Referee Comment (RC3) · Anonymous Referee #3 · 9 Jun 2017

GENERAL COMMENTS

The authors are to be congratulated for undertaking this major body of work to produce a consistently processed dataset of such long duration stretching back more than fifty years. It is often very difficult to work with such old data and (if it still exists) metadata. The NOAA Dobson record is certainly a crucial dataset for science representing many regions of the globe over these decades, not just the USA. The authors are therefore also to be commended for documenting their reprocessing activity for the ongoing benefit of all users of the data.

[Figure]

However, in its current form I don't believe the paper is acceptable for ACP, due to its lack of the appropriate level of rigour, and of transparency, for a scientific publication. In too many places the reader unfortunately gets the impression that an old black box has been replaced by a new black box, the operation of both of which is left completely mysterious. At many of the stations shown, the daily values of ozone have very frequently changed by as much as 10-20%, this is a big difference ($\sim$ 50 Dobson Units) and needs a proper explanation if the user is to have any confidence in the new dataset, and to meet modern expectations of transparency of data processing.

The revised version of the manuscript should include specific explanation of the old processing as far as possible, but much more importantly, proper explanation of what the current software (WinDobson) is doing. Without this, the current paper cannot serve as any sort of documentation of the resubmitted data in the WOUDC and NDACC databases. Certainly, the single document referred to with regard to WinDobson is not at all adequate as it contains no information at all about what the software actually does to the data. The major issues of concern to me in this respect are: (1) How WinDobson analyses an intercomparison to deduce the calibration – this seems to be different to the old system? (2) The difference between the "statistical methods" used to calculate zenith corrections in the old and new systems. (The results in some places differ substantially for an unknown reason, for example, refer lines 217-218). (3) The different methods for selection of a representative daily value. (There is discussion of the complications of time zones but I can't find a clear statement of how WinDobson does this selection).

Another general comment is that it seems in many cases (for a reason that is obscure to me) the WOUDC archive is missing long periods of data (indeed whole years at many stations eg Mauna Loa, South Pole, Boulder) and at some other stations WOUDC holds an out-of-date version of the data (eg Wallops Island, OHP, Perth, Lauder). Perhaps it would be more pertinent to compare the new dataset with the internal NOAA archive in these cases, which I assume doesn't contain these long gaps and has the most recent

re-processing results? (This is only a suggestion.)

SPECIFIC COMMENTS Line 20: "data quality controls built into the new software" – I am not saying this should be in the abstract but in the body of the paper, the authors should explain it what these tools are, and in particular, is the software identifying or removing bad data (and if so, how?) or are the tools merely GUIs to assist manual QC?

Line 29 "either . . . and " should be "either . . . or"

Line 29 Does the figure really add anything? It seems to be based heavily on WMO Report GAW 183? The optical arrangement seems to be included just for interest rather than being referred to again in the text.

Line 33: "The importance of the Dobson Spectrophotometer and its measurements are demonstrated by use of Dobson Units . . . " This statement does not follow logically and is not suitable for a scientific publication.

Line 33 "KM" should be "km"

Line 38 – GAW Report 183 should be listed in the references.

Line 40 "The instrument's readings . . . caused by its passage" – the way this sentence is written the instrument is passing through the ozone layer!!

Line 41 "N-value" should be better explained

Line 41 I object to the use of the terminology "RtoN tables". The community term is "N tables", I think this paper needs to be consistent with GAW Report 183 and all earlier reports and papers and not introduce different terms for the same thing.

Line 43 "The usefulness" – "usefulness" is not the right word here. The table will always be useful but it might become inaccurate over time.

Lines 45-47 The various metrological terms are not being used in the ISO sense as

recommended by GCOS (eg GCOS 200 page 293), but I concede many in the scientific community do not follow these either.

Line 46 – where does the figure of 1% come from? Is there a source for this?

Line 47 – "the accuracy is dependent on knowledge of the ozone and temperature profile" – I find this wording misleading because in fact you don't have any knowledge of the ozone or temperature profile at the time of measurement and have to make assumptions

Line 48 "a static value" – I also find this wording misleading because a reader might assume each station has its own (static) value – I suggest re-wording to make it clearer that the same value is used at all locations whatever their geographic position as well as all times of the year. The fact that the height of the ozone layer also is just approximated should also be stated.

Line 52 – give a source for the "2-5%"

Line 61 – "there are measurements of TOC" – perhaps change to "there are records of measurements .."

Line 65 – "Two stations have been either closed or been transferred .." – wouldn't it be easier to say one station has been closed and one has been transferred?

Line 77 – "RLA" – I don't believe this term is commonly used in the Dobson community but I might be wrong – I can't find it in GAW #183.

Line 79 "referencing" -> referencings

Line 82 "calibration tables" – this term needs defining – the reader will not know if it is the same as the N-tables or not.

Line 84 "is to be reprocessed" – this is a crucial point that needs to be explicit. Is the correction applied backwards in time either by the new system or the old system, and if so, how?

Line 84 "is to be reprocessed" – I find this expression strange, it sounds like something from a manual, but the paper needs to say what has actually been done in practice.

Line 85 – ". . . by the 2010s were difficult to use and maintain" – perhaps this comment is more relevant to the work itself and not so relevant to the paper, but I don't find it very credible. It couldn't have been too difficult to recompile the old fortran code on a modern PC (unless maybe the source code had been lost?) I am sure the old programs couldn't have been very complicated.

Line 89 – the document at the weblink is really just an advertising brochure containing various screen shots. It has no information about what the software actually does in terms of how it treats the data. I see this as the single major weakness of the present manuscript. I don't believe it is acceptable in the year 2017 to submit data to databases but without disclosing how the data have been processed.

Line 90 – "this software has a different statistical method . . . and set of rules . . . " - which need to be explained in the next section

Line 98 "personnel inspection" – perhaps replace with "human inspection" or "inspection by personnel"?

Line 103 ". . .comparisons . . . could be performed using tools internal to WinDobson . . ." – the important thing here which is left unsaid is what was done with the comparison values? Was it just for interest or was it part of the QC process? Does WinDobson automatically exclude outliers? Clearly if you're deleting different points in a day that will change the daily value unless I am missing something?

Line 106 "fundamental wavelength pairs" – there has been no explanation of why AD-DS are being considered "fundamental"?

Line 108 "Time periods with differences greater than this were investigated to determine the source of the problem, and correct any differences" I can't understand what you were doing here, sorry, were the differences caused by mistakes of some sort?

Please clarify.

Line 113 "The new method has resulted in ∼91% of zenith sky derived total ozone (ADZB) within 2% . . ." Is this comparison based on the same time period that has been used to derive the coefficients for the statistical relationship, or have you used one period to calculate the coefficients and then a second period to test the fit? Otherwise it is possible to over-fit an overly complicated function (eg 6-degree polynomials in multiple variables) which gives excellent results in the training period but not afterwards.

Line 114 "the 2006 Operations Handbook" – this report should be referenced and referred to by a consistent name.

Line 114 "the 78% value" – but this value comes from a short study conducted in the 1950s!! Surely there is something more recent and thorough you could compare to?

Line 128 – ". . . some adjustments were made in the WinDobson process for some stations" – please explain what you did – this seems very arbitrary?

Line 133 "The older processing included time periods of special processing . . ." – maybe I am misunderstanding this, but it sounds like previously attempts were made to correct for the two mentioned problems, but now you're not going to try to correct for them anymore? Why wouldn't this be a problem?

Line 140 "The older processing modified the reference lamp correction . . . " I just can't make sense of this sentence sorry, is it possible to make it clearer?

Line 149 "For some stations . . ." The paragraph explains why this is tricky, but I can't see any clear statement of how it should be done, or how the old software used to do it, or how the new software does it? This needs to be explicit.

Line 154 "Data archives sometimes failed to be updated . . ." I also found this statement hard to make sense of. Are you saying NOAA updated your internal records but neglected to pass on the reprocessed data to WOUDC?

Line 168 "the new R-N tables" – this is inconsistent with the terminology used earlier in the manuscript "RtoN tables" but again I would prefer the community-accepted term "N tables".

Line 174 ". . . probability distributions . . . " I agree distributions are the clearest way to show the difference. The fact the curves are symmetric shows there is no systematic bias but I would have thought the most important point was the width of the curves reflecting the uncertainty.

Line 183 ". . .compared to WinDobson record" -> ". . . compared to the WinDobson record"

Line 189 "The NDACC archive appears to have updates not reflected in the WOUDC Archive." – How is this situation possible? Did NOAA forget to send in the reprocessed data or was it a deliberate decision? Does NOAA retain its own archive which contains all updates?

Line 200 "There are several periods missing from the archive, including all of 2015". Again, I don't really understand this situation? Why would a whole year be missing? Given the large apparent gaps, would this study be more meaningful if it compared NOAA's internal archive rather than WOUDC?

Line 217 "This station record shows a larger offset . . .. Due to the change in zenith observation results" But why? What exactly has changed? Why would it be bigger at this station than the other? Wouldn't Nashville be sunny anyway and have a smaller proportion of zenith observations?

Line 227 "The selection of observations should be changed . . ." What does "should be" mean here? Have they actually been changed in the new processing or not?

Line 237 "The data from July 2013 to July 2014 is missing from the Archive". I assume "the Archive" means WOUDC? Again we have the situation I find very strange that a whole year is simply not present.

Line 246 "Alaska" -> "Alaska, USA"

Figure 2 – I think this is a good plot but:

- South Pole seems to be missing.

- The blue lines for "others" seem pretty bad to me at some stations, eg BNA, FBK, WAI, BRW, SMO. This does not seem consistent with the earlier claim of 2-5% for zenith readings.

Figure 3 – I think this is a good plot too.

Figures 4-17 general comments

- In the caption, I would prefer the full station name be given rather than the 3-letter code, a reader outside of NOAA would find this cumbersome

- Does panel 1 really show daily values? There don't seem to be enough dots.

- It is confusing enough, that panel 1 shows ozone in DU but then panel 2 changes to percentage difference, but made doubly so by the fact that the y-axes aren't labelled!

- In panels 2 and 4, rather than the red line showing the linear trend, which I don't think is very pertinent, it would be better to show the zero line

- Panels 3 and 5 should also show a zero line

- Do the labels on the black vertical lines mean the end of each year? Usually "2015" on a tick mark would label the start of 2015, not the end.

Figure 5 – There is an abrupt shift in the mid 1980s which looks unphysical – could you comment on this?

- Panels 4 and 5 show some very high values for the differences, many months being between 10 and 20%. Can you really account for this?

Figure 11

- In the first ten years or so there are some very low values of total ozone (down to 200 DU) which then disappear after 1979. This looks like bad data to me.

- There is a step change in the difference around 2005 but I didn't see any explanation for the cause.

Table 2 I'm not sure the offset and, in particular the linear trend, are worth giving in the table. It would seem very unlikely that a reprocessing such as this would end up resulting in a long-term trend. I would rather see a summary of the distributions shown in figure 2, such as 2 sigma values for, perhaps AD-DS, CD-DS, AD-ZB, AD-ZC .

Table 3 I think the idea of this table is good but it is slightly misleading because there seems to be a lot of variations between the stations and the table shows combined results. Some of the stations have much greater spread than the overall average figures. However, I wouldn't object to this if table 2 could be changed to give station-by-station distribution figures as suggested above.

---

## Author Response (AR1)

We thank the reviewers for the time and helpful comments on the manuscript. Some comments were common to all reviews and we address these first before addressing more specific comments to each reviewer.

*A common note was that the individual station plots lacked units and were difficult to understand. The plots have been redone for the revised manuscript. The conclusion was expanded, and details were added to the revised manuscript.*

Reviewer 1 comments

This reviewer has several issues with this paper but all are easily remedied.

1) The primary issue is that the data are not publically available so the reviewer(s) are not able to check any claims made by the authors. The policies of ACPD are a bit vague but many publications will not allow submission of papers citing proprietary data. I suggest putting the new data somewhere where it can be accessed before the release of this paper.

> *Response. We have an ftp site for the retrieval of the daily values from the WinDobson processing. The actual WinDobson software will be made available on request to the authors. The following sentence is added to the abstract: "The new WinDobson data is now available from 'ftp://aftp.cmdl.noaa.gov/data/ozwv/Dobson/WinDobson/. The WOUDC archive is available from http://woudc.org/, and the NDACC archive is available from ftp://ftp.cpc.ncep.noaa.gov/ndacc/ The WinDobson Software, Level-0 Dobson data and calibration records are available on request to NOAA"*

2) The second issue is concerning the conclusion section. This section is extremely underwhelming. The reader really wants to know WHY this work was done and to how the Dobson data has changed in a scientific sense. There are large differences in the data at high latitudes (+/- 10 percent!) which should be very easily studied. A couple of paragraphs statistically comparing the new & old datasets to the satellite overpasses for those Dobson stations would be highly useful as would a plot or two, and that added work would make this paper much stronger and more complete.

> *Response. The conclusion has been rewritten. A formal comparison between these ground stations and the various satellite data records is planned as a separate publication.*

3) Can you quantify how much better the new data are? Less noise? Fewer step functions? Less bias? This is the payoff for all that hard work analyzing and revising the old data.

> *Response. This issue is addressed in the rewritten conclusion section*

> *The overall changes are small (~0.1% offset), but several individual stations have a larger offset (Maximum 0.7%) driven by the changes in the ZC reduction polynomials. With the comparisons with the existing NDACC and WOUDC archives, we were able identify periods with either missing data or incorrectly processed data. The differences between the old and the planned updated archives have overall small offsets and trends (Table 2), but within the long-term record that are*

*periods with greater differences of which researchers should be aware (see figures 4 through 16, and description of the individual station histories.). The paper includes a section that describes individual station histories which provides information on specific to station updates and their effects on the total ozone record. The offsets and trends for differences between the old and the new version of the data are not the same for WOUDC and NDACC archives, as the NDACC set of data is not a perfect match to the one available from the WOUDC archive . For example, Wallops Island NDACC record is 1995-2014, while the WOUDC record is 1967-2014. When the NDACC and WOUDC archives are updated, these archived datasets will be complete and homogenized. Moreover, after all calibrations and the applicable periods were reviewed, the history method of applying calibrations to all of the instruments in the networks has been standardized. The new WinDobson database, available to researchers on request, will allow investigators improving the accuracy of the Dobson retrieval algorithms.*

4) Issue 3: When a new calibration or instrument repair was done how were the **new calibration values applied**? Were they put in as a **step function or gradually introduced over time** (linearly?) between known calibrations/changes?? Please explain.

*Response. We have put a more complete description of the process in the manuscript. "Our investigations of the station and instrument operation history revealed several periods for which different N-tables were used in the archived records as compared to the historical record of NOAA N-tables. Also, when a station instrument is compared to a standard instrument, and the results are within the uncertainty of the measurements (+/-1%), the station instrument's calibration is considered to be stable and thus is not changed. Otherwise, the instrument's calibration is changed and the existing data record starting from the time of the last comparison against a standard is reprocessed with the assumption that instrument's calibration has changed in a linear manner. Using the tools in WinDobson, our studies of the stations' records allowed comparisons with long term records indicating TOC. These comparisons showed that at certain stations, the calibration change was not linear. Further investigation of stations' history revealed damage to the instrument at that point (for example, rain entering instrument shelter.) These investigations also identified instances where the comparison against the Dobson standard was not performed correctly, and therefore the calibration should not have been changed."*

5) Specific little changes recommended:

Line 16: remove "for possible changes"

*Response. We have fixed the following issues in the revised manuscript.*

Line 25: remove comma

I recommend putting lines 36-8 after line 59

Line 57 & 80: you may want to define what the optical wedge is.

Figure 3 is referenced before Figure 2.

Please fix Line 149 is not a sentence

Line 157 selected value of what?

Line 182&235: Change was to were

Line 207: And 'the' before Bismarck

Line 236: use "There are data in the archive prior to 1966 but are not connected. . ..

Line 237: change is to are

Line 242: remove commas

Line 247: remove comma

Line 293: "station" appears twice.

All plots are daily averaged Dobson values or individual measurements? Please put units on plots!

*Response: The points are daily selected values.  Plots have been redone with units, and more information.*

Anonymous Referee #2

General Comments:

This technical note presents the re-evaluation of the total ozone data record derived by Dobson spectroradiometers operating by NOAA. The reprocessed data are com- pared with the data already deposited to the databases of WOUDC and NDACC. The manuscript includes important information from the history of the different stations and the problems encountered during their long term operation. I think it is a good practice to

publish such information which is usually accessible only from the stations' personnel.

I see two main weaknesses in this paper. First, the WinDobson software package is not described adequately so that the reader cannot assess the differences in processing of the data compared to the traditional methods. The link provided as reference (end of page 3) does not help because it is a very brief slide presentation. Please, either provide a more suitable reference where the methods are described in detail, or include more details in the text.

*Response: This text was added "The new WinDobson data is now available from ftp://aftp.cmdl.noaa.gov/data/ozwv/Dobson/WinDobson/. The WOUDC archive is available from http://woudc.org/, and the NDACC archive is available from ftp://ftp.cpc.ncep.noaa.gov/ndacc/ The WinDobson Software as extended by NOAA, Level-0 Dobson data and calibration records are available on request from NOAA Dobson network personnel https://www.esrl.noaa.gov/gmd/ozwv/dobson/contact.html"*

*Also: "Developed by personnel of the Japan Meteorological Agency (Miyagawa, 1996), WinDobson is a software package for operations, data analysis and quality assurance of Dobson spectrophotometer observations. The algorithm for the reduction of ozone from DS observations with the Dobson is the standard method used by the NOAA software, but the ZS observations are reduced with a method described later in this manuscript.*

Second, as the authors have long experience and deep knowledge of the Dobson retrieval algorithms and the instrument details, they tend to present their thoughts very briefly assuming that the reader has the same level of knowledge. I suggest to provide some more details so that even people who are not involved directly in the Dobson measurements can follow the paper easily.

*Response: We have added more details in a revised manuscript.*

Finally there are some rather minor presentation problems that are mentioned explicitly in my specific comments.

I suggest to accept this technical note for publication in ACP after taking into account the following suggestions in the revised manuscript:

Specific comments by line number:

2: Please change to "… intensity of solar radiation between …"

*Response: Updated*

32-34: Something is missing in this sentence. The importance of the Dobson could not be demonstrated by "using Dobson units".

*Response: This sentence has been removed*

38: This handbook should be included in the References section.

*Response: Updated*

41: Please insert here a reference as the meaning of the R and N values is understandable only from scientists experienced in the Dobson.

*Response: Addressed by adding more detail in rewording the previous paragraph. This sentence is changed to "The relative intensity of a wavelength pair outside the Earth's atmosphere is referred to as the extraterrestrial constant (ETC), and the Dobson spectrometer exploits the change in that relationship caused by the passage of UV light through the ozone layer. This is performed by passing a neutral density filter (the optical "wedge") across the light path of the wavelength less absorbed by ozone--the light passing through slit S-3 (figure 1). The instrument's output, R-value, indicates the position of the neutral density filter as indicated on an engraved plate, and an N- value is the corresponding attenuation caused by the neutral density filter at that position combined with the instrument's ETC. The instrument 's ETC is determined either through a Langley Plot method or by direct comparison with a standard Dobson instrument."*

43: Would "applicability" be a better choice than "usefulness"?

*Response: Agreed and Updated.*

73: Although traditionally ozone was archived as single daily values, nowadays individual values within a day are also available. Please revise this sentence accordingly.

*Response:  Sentence changed. "TOC is normally archived as a single representative value of TOC selected for each day. This not an average value, but the result from the "best" observation during the day. As the exact instrumentation and observational scheduling varies from station to station, the number of observations made daily also vary. The full record of observations is available per request from NOAA Dobson network personnel listed at https://www.esrl.noaa.gov/gmd/ozwv/dobson/contact.html"*

150Anonymous Referee #3

GENERAL COMMENTS

The authors are to be congratulated for undertaking this major body of work to produce a consistently processed dataset of such long duration stretching back more than fifty years. It is often very difficult to work with such old data and (if it still exists) metadata. The NOAA Dobson record is certainly a crucial dataset for science representing many regions of the globe over these decades, not just the USA. The authors are therefore also to be commended for documenting their reprocessing activity for the ongoing benefit of all users of the data.

However, in its current form I don't believe the paper is acceptable for ACP, due to its lack of the appropriate level of rigor, and of transparency, for a scientific publication. In too many places the reader unfortunately gets the impression that an old black box has been replaced by a new black box, the operation of both of which is left completely mysterious. At many of the stations shown, the daily values of ozone have very frequently changed by as much as 10-20%, this is a big difference ($\sim$ 50 Dobson Units) and needs a proper explanation if the user is to have any confidence in the new dataset, and to meet modern expectations of transparency of data processing.

> *Comment: This paper is designed to present the new WinDobson dataset to the past Dobson users, to inform them of the changes and improvements in the new datasets. Most of the changes in the new NOAA Dobson total ozone datasets are brought by the conversion from the old processing system to the new, and ACP's special issue regarding NDACC anniversary seem to be the correct platform to discuss upcoming changes in the reprocessed NOAA Dobson datasets. We acknowledge that the old system had become a "black box" due to changes of personnel and inconsistent methods used through the historical record to create Dobson record. We hope that our revised manuscript meets your requirements for publication.*

The revised version of the manuscript should include specific explanation of the old processing as far as possible, but much more importantly, proper explanation of what the current software (WinDobson) is doing. Without this, the current paper cannot serve as any sort of documentation of the re-submitted data in the WOUDC and NDACC databases.

Certainly, the single document referred to with regard to WinDobson is not at all adequate as it contains no information at all about what the software actually does to the data.

The major issues of concern to me in this respect are:

(1) How WinDobson analyses an intercomparison to deduce the calibration – this seems to be different to the old system?

*Response: The mathematical analysis of the intercomparison between a standard and "test" instrument in WinDobson is- based on the NOAA version. WinDobson adds more visual information to assist with interpretation of the results. In either system, expert human input determines the final calibration, and application of this information to the existing record.*

(2) The difference between the "statistical methods" used to calculate zenith corrections in the old and new systems. (The results in some places differ substantially for an unknown reason, for example, refer lines 217-218).

*Response: The methods of reducing zenith observations have evolved from empirical hand-drawn charts, to digitized charts converted to table look-up algorithms, and now to multi-variable polynomials. The older methods are difficult to update. The polynomial method can be selectively applied to specific time periods in the record. In regards to the station referred to in lines 217-218, the algorithm designed in 1995 was likely incorrect.*

(3) The different methods for selection of a representative daily value. (There is discussion of the complications of time zones but I can't find a clear statement of how WinDobson does this selection).

*Response: Briefly, in both NOAA and WinDobson system, the mostly likely selected observation is an ADDS observation near local noon. WinDobson has some different "weighting" in the selection. A section was added:*

*"Windobson Selection Rules.*

*Often there are multiple observations on an individual day. The observations are given an internal numeric code in WinDobson, based on the observation type, and operator input about the observation atmospheric conditions. The representative value is chosen by the software with the priority groups given below, high to lowest. If there are multiple observations of the highest priority on that day, the observation closest in time to local noon is chosen. After the automatic selection, the daily representative values are reviewed by human inspection with possible intervention to select a different value. The WinDobson software also has quality control routines that rates individual observations as good, questionable (flagged yellow) and likely bad (flagged red), based on internal consistencies of the measurements. If an observation is rejected by the human inspector, the observation is not removed from the data record, but flagged as "not included".*

*Priority Groups are listed here; Operator inputs as to sky quality are included in determining priority:*

*1. Direct Sun observations using the AD pair combination with or without Ground Quartz Plate*

*(diffuser) in the instrument's inlet window. Observations with diffuser have higher priority.*

2. *Zenith Sky observations using the AD pair combination, observations on the clear zenith have higher priority over those on cloudy conditions*
3. *Direct Sun observations using the CD pair combination with Ground Quartz Plate (diffuser) in the instrument's inlet window. Observations without diffuser have lower priority.*
4. *Zenith Sky observations using the CD pair combination, observations on the clear zenith have higher priority over those on cloudy conditions.*
5. *Zenith Sky observations using the CC' pair combination, observations on the clear zenith have higher priority over those on cloudy conditions*
6. *Observations on light reflected from the moon. Observations using AD pair combination have higher priority. Note these observations are rarely made other than at the South Pole Station during the austral winter."*

Another general comment is that it seems in many cases (for a reason that is obscure to me) the WOUDC archive is missing long periods of data (indeed whole years at many stations eg Mauna Loa, South Pole, Boulder) and at some other stations WOUDC holds an out-of-date version of the data (eg Wallops Island, OHP, Perth, Lauder). Perhaps it would be more pertinent to compare the new dataset with the internal NOAA archive in these cases, which I assume doesn't contain these long gaps and has the most recent re-processing results? (This is only a suggestion.)

> **Response:** *The general portals for the daily Total ozone values are the WOUDC and NDACC archives. It is true that many researchers have requested observations for certain stations and periods directly from NOAA. The historic internal NOAA archive will be retained, but the WinDobson archive will become the operation archive.*

SPECIFIC COMMENTS

Line 20: "data quality controls built into the new software" – I am not saying this should be in the abstract but in the body of the paper, the authors should explain it what these tools are, and in particular, is the software identifying or removing bad data (and if so, how?) or are the tools merely GUIs to assist manual QC?

> **Response:** *We have added various details throughout the revised manuscript.*

Line 29 "either . . . and " should be "either . . . or"

> **Response:** *Corrected*

Line 29 Does the figure really add anything? It seems to be based heavily on WMO Report GAW 183? The optical arrangement seems to be included just for interest rather than being referred to again in the text.

245      *Response: The revised section "Background" know refers to the details in figure 1.*

Line 33: "The importance of the Dobson Spectrophotometer and its measurements are demonstrated by use of Dobson Units . . . " This statement does not follow logically and is not suitable for a scientific publication.

     *Response: Sentence removed.*

Line 33 "KM" should be "km" Line 38 – GAW Report 183 should be listed in the references.

250      *Response:  The lines 25-59 have (Section: Background) have been revised with more detail.*

Line 40 "The instrument's readings . . . caused by its passage" – the way this sentence is written the instrument is passing through the ozone layer!!

     *Response:  The lines 25-59 have (Section: Background) have been revised with more detail.*

Line 41 "N-value" should be better explained

255      *Response:  The lines 25-59 have (Section: Background) have been revised with more detail.*

Line 41 I object to the use of the terminology "RtoN tables". The community term is "N tables", I think this paper needs to be consistent with GAW Report 183 and all earlier reports and papers and not introduce different terms for the same thing.

     *Response: Agreed and Updated*

260 Line 43 "The usefulness" – "usefulness" is not the right word here. The table will always be useful but it might become inaccurate over time.

     *Response:  The lines 25-59 have (Section: Background) have been revised with more detail.*

Lines 45-47 The various metrological terms are not being used in the ISO sense as recommended by GCOS (eg GCOS 200 page 293), but I concede many in the scientific community do not follow these either.

265      *Response:  The word "uncertainty" has been added*

Line 46 – where does the figure of 1% come from? Is there a source for this?

     *Response:  The lines 25-59 have (Section: Background) have be revised with more detail.*

Line 47 – "the accuracy is dependent on knowledge of the ozone and temperature profile" – I find this wording misleading because in fact you don't have any knowledge of the ozone or temperature profile at the time of 270 measurement and have to make assumptions

*Response:* *The lines 25-59 have (Section: Background) have been revised with more detail.*

Line 48 "a static value" – I also find this wording misleading because a reader might assume each station has its own (static) value – I suggest re-wording to make it clearer that the same value is used at all locations whatever their geographic position as well as all times of the year. The fact that the height of the ozone layer also is just approximated should also be stated.

*Response:* *The lines 25-59 have (Section: Background) have been revised with more detail.*

Line 52 – give a source for the "2-5%"

*Response:* *The lines 25-59 have (Section: Background) have been revised with more detail.*

Line 61 – "there are measurements of TOC" – perhaps change to "there are records of measurements .."

*Response: Agreed and Updated*

Line 65 – "Two stations have been either closed or been transferred .." – wouldn't it be easier to say one station has been closed and one has been transferred?

*Response: Agreed and Updated*

Line 77 – "RLA" – I don't believe this term is commonly used in the Dobson community but I might be wrong – I can't find it in GAW #183.

*Response: Agreed and removed this term.*

Line 79 "referencing" -> referencings

*Response: Agreed and Updated*

Line 82 "calibration tables" – this term needs defining – the reader will not know if it is the same as the N-tables or not.

*Response: Agreed and Updated*

Line 84 "is to be reprocessed" – I find this expression strange, it sounds like something from a manual, but the paper needs to say what has actually been done in practice.

*Response: Sentence changed to "When the calibration N-tables are changed due to a drift (determined from an inspection of the past calibrations, instrument operational history, and, if possible, comparison with other instrumental records), the existing data set from the last calibration change to the new calibration was reprocessed and re-published in the archives."*

Line 85 – ". . . by the 2010s were difficult to use and maintain" – perhaps this comment is more relevant to the work itself and not so relevant to the paper, but I don't find it very credible. It couldn't have been too difficult to recompile the old Fortran code on a modern PC (unless maybe the source code had been lost?) I am sure the old programs couldn't have been very complicated.

*Response: This is a NOAA workforce issue, and the reason for change in the processing software.*

Line 89 – the document at the weblink is really just an advertising brochure containing various screen shots. It has no information about what the software actually does in terms of how it treats the data. I see this as the single major weakness of the present manuscript. I don't believe it is acceptable in the year 2017 to submit data to databases but without disclosing how the data have been processed.

*Response: We have removed this sentence and modified the following to read: "Developed by personnel of the Japan Meteorological Agency (Miyagawa, 1996), WinDobson is a software package for operations, data analysis and quality assurance of Dobson spectrophotometer observations. For the NOAA application, new components were developed. These new components are available from NOAA to other users of WinDobson. It is applicable for both TOC and Umkehr (ozone vertical profile) measurements. As this software has a different statistical method for the reduction of the zenith measurements, and set of rules (See section: Windobson Selection Rules) for determining the representative value of total ozone for each day with observations, the entire data record of each operational station was reprocessed in the WinDobson system to minimize the effect of the change when future data is placed in the archive In the development of the data files and calibration information for Windobson processing, the entire record of observations, repair and calibration checks of each station was investigated and re-evaluated. This investigation allows for correction of past errors.*

Line 90 – "this software has a different statistical method . . . and set of rules . . . " - which need to be explained in the next section

*Response: This has been changed with the response to previous comment*

Line 98 "personnel inspection" – perhaps replace with "human inspection" or "inspection by personnel"?

**Response:** Agreed and Updated

Line 103 ". . .comparisons . . . could be performed using tools internal to WinDobson . . ." – the important thing here which is left unsaid is what was done with the comparison values? Was it just for interest or was it part of the QC process? Does WinDobson automatically exclude outliers? Clearly if you're deleting different points in a day that will change the daily value unless I am missing something?

*Response: The initial comparison with external records was done to identify errors in the selection of*

330 *appropriate N-tables, a part of the QC Process.  WinDobson does not automatically exclude outliers, but does choose a single observation as the representative daily value.  This observation may not be the same as the one chosen in the NOAA processing. This can be changed- by inspection by personnel.  We have added more details.*

Line 106 "fundamental wavelength pairs" – there has been no explanation of why ADDS are being considered
335 "fundamental"?

> **Response:** *We added the following sentence: "The ADDS observations are considered the most reliable (fundamental), as the equation derived for conversion to ozone minimizes the Rayleigh scattering term, and the aerosol term can be considered to be zero."*

Line 108 "Time periods with differences greater than this were investigated to determine the source of the
340 problem, and correct any differences" I can't understand what you were doing here, sorry, were the differences caused by mistakes of some sort?  Please clarify.

> **Response:** *The sentence was poorly written and will be replaced.  "Time periods with differences greater than this +/-1 DU were investigated to determine the source of the problem, and correct any differences."*

345 Line 113 "The new method has resulted in ~91% of zenith sky derived total ozone (ADZB) within 2% . . ." Is this comparison based on the same time period that has been used to derive the coefficients for the statistical relationship, or have you used one period to calculate the coefficients and then a second period to test the fit? Otherwise it is possible to over-fit an overly complicated function (eg 6-degree polynomials in multiple variables) which gives excellent results in the training period but not afterwards.

350 > **Response:** *The polynomial coefficients are defined for a particular time periods, using days with both ADDS and ZS observations.  The 91% number is based on that same time period.*

Line 114 "the 2006 Operations Handbook" – this report should be referenced and referred to by a consistent name.

> **Response:** *Agreed and Updated*

355 Line 114 "the 78% value" – but this value comes from a short study conducted in the 1950s!! Surely there is something more recent and thorough you could compare to?

> **Response:** *The NOAA processing was derived from the study in the 1950s, so this comparison is appropriate.*

Line 128 – ". . . some adjustments were made in the WinDobson process for some stations" – please explain

360    what you did – this seems very arbitrary?

> *Response: Changed to "The history of the instrument calibrations was again reviewed, and changes in the N-tables and the periods of the use of N-tables within the WinDobson system were made as needed. The differences stem from a number of reasons."*

Line 133 "The older processing included time periods of special processing . . ." – maybe I am misunderstanding
365    this, but it sounds like previously attempts were made to correct for the two mentioned problems, but now you're not going to try to correct for them anymore? Why wouldn't this be a problem?

> *Response: We are changing the text of this section to better explain how the older (1995 and earlier) special processing was imported into WinDobson (extracting the adjustments from the LLF files). The method for applying drift corrections is different in WinDobson processing, and no attempt is made to
370    "correct" for SZA dependency with some instruments, as the dependency is controlled by the internally scattered light within the instrument. This internally scattered light has yet to be measured for individual Dobson instruments.*

Line 140 "The older processing modified the reference lamp correction . . . " I just can't make sense of this sentence sorry, is it possible to make it clearer? Line 149 "For some stations . . ." The paragraph explains why
375    this is tricky, but I can't see any clear statement of how it should be done, or how the old software used to do it, or how the new software does it? This needs to be explicit.

> *Response: We have updated the text of this section for a clearer explanation. "*

Line 154 "Data archives sometimes failed to be updated . . ." I also found this statement hard to make sense of. Are you saying NOAA updated your internal records but neglected to pass on the reprocessed data to WOUDC?

380    *Response: There are times when either the data was not transmitted to the WOUDC, or the WOUDC did not update the archive. This work will correct those mistakes.*

Line 168 "the new R-N tables" – this is inconsistent with the terminology used earlier in the manuscript "RtoN tables" but again I would prefer the community-accepted term "N tables".

> *Response: Agreed and Updated*

385    Line 174 ". . . probability distributions . . . " I agree distributions are the clearest way to show the difference. The fact the curves are symmetric shows there is no systematic bias but I would have thought the most important point was the width of the curves reflecting the uncertainty.

> *Response: Agreed and Updated*

Line 183 ". . .compared to WinDobson record" -> ". . . compared to the WinDobson record"

390       *Response: Agreed and Updated*

Line 189 "The NDACC archive appears to have updates not reflected in the WOUDC Archive." – How is this situation possible? Did NOAA forget to send in the reprocessed data or was it a deliberate decision? Does NOAA retain its own archive which contains all updates?

      *Response: There are times when either the data was not transmitted to the WOUDC, or the WOUDC did*
395       *not update the archive.   This work will correct those mistakes.*

 Line 200 "There are several periods missing from the archive, including all of 2015". Again, I don't really understand this situation? Why would a whole year be missing? Given the large apparent gaps, would this study be more meaningful if it compared NOAA's internal archive rather than WOUDC?

      *Response:  Many researchers use the WOUDC archive, and need to understand that the time series in*
400       *the archive has changed.*

Line 217 "This station record shows a larger offset . . .. Due to the change in zenith observation results" But why? What exactly has changed? Why would it be bigger at this station than the other? Wouldn't Nashville be sunny anyway and have a smaller proportion of zenith observations?

      *Response:  Nashville site is the National Weather Office near Old Hickory, TN.  The change to the*
405       *polynomial method for reduction of zenith observations changed the results of these observations*
       *enough to create an offset, as 43% days are reported as ZS.  Boulder reports 28% ZS.*

Line 227 "The selection of observations should be changed . . ." What does "should be" mean here? Have they actually been changed in the new processing or not?

      *Response:  Sentence is changed. "Researchers are advised that this instrument shows patterns in the*
410       *comparison with other instrumentation that imply an under estimation of ozone on the ADDS*
       *wavelength under conditions of low sun and high ozone."*

 Line 237 "The data from July 2013 to July 2014 is missing from the Archive". I assume "the Archive" means WOUDC? Again we have the situation I find very strange that a whole year is simply not present.

      *Response:  The archiving will correct these errors.*

415  Line 246 "Alaska" -> "Alaska, USA"

      *Response:  Fixed.*

Figure 2 – I think this is a good plot but: - South Pole seems to be missing. - The blue lines for "others" seem pretty bad to me at some stations, eg BNA, FBK, WAI, BRW, SMO. This does not seem consistent with the earlier

claim of 2-5% for zenith readings.

420     *Response:  This plot is updated with correct naming. The blue line for others (ZS) is the probability of a change in the zenith result with the change to*

Figure 3 – I think this is a good plot too.

 Figures 4-17 general comments - In the caption, I would prefer the full station name be given rather than the 3-letter code, a reader outside of NOAA would find this cumbersome - Does panel 1 really show daily values?
425     There don't seem to be enough dots. - It is confusing enough, that panel 1 shows ozone in DU but then panel 2 changes to percentage difference, but made doubly so by the fact that the y-axes aren't labelled! - In panels 2 and 4, rather than the red line showing the linear trend, which I don't think is very pertinent, it would be better to show the zero line - Panels 3 and 5 should also show a zero line - Do the labels on the black vertical lines mean the end of each year? Usually "2015" on a tick mark would label the start of 2015, not the end.

430     *Response:  The Plots and captions have been updated with more information.  Panel 1 shows each day that has a selected value reported.  Not all stations are operated seven days a week.*

 Figure 5 – There is an abrupt shift in the mid 1980s which looks unphysical – could you comment on this? - Panels 4 and 5 show some very high values for the differences, many months being between 10 and 20%. Can you really account for this?

435     *Response:  This is the South Pole station.  The station history paragraph is changed to "*

*South Pole Station was established in 1957. The first Dobson instrument failed due to the extreme cold. Observations started again in 1961 and these results are in the NOAA archive, but the calibration record dates from 1963.  The normal routine established in 1985 is to change the instrument every four years for calibration checks, but this was not always achieved.  This station has the possibility of large changes in reported daily*
440     *values in the WinDobson, primarily due to the extended daily observation period, and high variation in total ozone during certain periods of the year. The station local day is the same as that of Christchurch, New Zealand for ease of logistics, but the Dobson observations are reported in the WOUDC in UTC date and hour. The date and time combination often is misleading (for example, In the WOUDC archive, 14 November 1994 has a time of 28 hours UTC, which matches the WinDobson and NDACC 15 November 1994 values.)   The calculation of the*
445     *astronomical parameters used in the algorithm for reducing reflected moon observations was incorrect in the NOAA program throughout the period of record.  Changes in the method of deriving total ozone from ZS observations improved the average with respect to DS averages, but creates differences between the old and new archives.   There are several periods missing from the WOUDC and NDACC archives (for example, July through December 2002.).  The difference between the WOUDC and NDACC archives records processed in the*
450     *NOAA system and WinDobson system are presented graphically in Figure 5. The exclusion of low TOC values in early October in the archived data (small white circles are outside of the plot range) in some years also produces*

*large percentage differences in the averages (see large deviations in open circles seen in some years in the panel c and e). An example is October 1994, where there are 25 reported days in the WinDobson record but only 18 reported in the WOUDC, and only 10 in the NDACC archive. These inconsistencies can produce large percentage*
455 *differences, especially during low ozone conditions.*

*The rules for selection and inclusion of days in the archives appear have been inconsistent in earlier (NOAA) processing and archiving. The NDACC archive prior to 1999 has TOC expressed as Vertical Column Density (molecules/cm\*\*2). These numbers appear to have been calculated from DU, as this archive is derived from the WOUDC archive. There are periods where this calculation was done incorrectly (for example, October 1998,*
460 *where the NDACC values differ by more than 100 DU when converted back to DU.) While the NDACC archive is supposed to be derived from the same internal NOAA archive as WOUDC, there are is random differences (For example, February 1981 is missing from the NDACC archive.) The change in the yearly cycle of TOC (Panel A) is evident in the austral spring due the depletion related to chlorofluorocarbon release (Farman etal, 1985). Station and observing schedules were changed to accommodate research needs after that 1985.*

465 Figure 11 - In the first ten years or so there are some very low values of total ozone (down to 200 DU) which then disappear after 1979. This looks like bad data to me. - There is a step change in the difference around 2005 but I didn't see any explanation for the cause.

    **Response: Thank you for noticing this error.  This plot is corrected.**

Table 2 I'm not sure the offset and, in particular the linear trend, are worth giving in the table. It would seem
470 very unlikely that a reprocessing such as this would end up resulting in a long-term trend. I would rather see a summary of the distributions shown in figure 2, such as 2 sigma values for, perhaps AD-DS, CD-DS, AD-ZB, AD-ZC
:

    *Response: We are retaining the table, as we think it useful.  The summary of figure 2 is given table 3. Note that the figure 3 gives the percent of ADDS results that change within +/-1% in conversion to*
475     *WinDobson.*

Table 3 I think the idea of this table is good but it is slightly misleading because there seems to be a lot of variations between the stations and the table shows combined results. Some of the stations have much greater spread than the overall average figures. However, I wouldn't object to this if table 2 could be changed to give station-by-station distribution figures as suggested above.

480     *Response: We are retaining the table, as we think it useful.*

[revised manuscript text omitted]